# Direct observation of selective autophagy induction in cells and tissues by self-assembled chiral nanodevice

Maozhong Sun[1,2], Tiantian Hao[1,2], Xiaoyun Li[3], Aihua Qu[1,2], Liguang Xu[1,2], Changlong Hao[1,2], Chuanlai Xu (ORCID) [1,2] & Hua Kuang[1,2]

The interactions between chiral nanomaterials and organisms are still challenging and mysterious. Here, a chiral nanodevice made of yolk–shell nanoparticles tetrahedron (UYTe), centralized with upconversion nanoparticles (UCNPs), was fabricated to induce autophagy in vivo. The proposed chiral nanodevice displayed a tunable circular dichroism (CD) signal when modified with different enantiomers of glutathione (GSH). Notably, UYTe showed significant chirality-dependent autophagy-inducing ability after D-GSH-modification because the enhanced oxidative stress and accumulation in living cell. The activation of autophagy resulted in the reduced intracellular CD intensity from the disassembly of the structure. The intracellular ATP concentration was simultaneously enhanced in response to autophagy activity, which was quantitatively bio-imaged with the upconversion luminescence (UCL) signal of the UCNP that escaped from UYTe. The autophagy effect induced in vivo by the chiral UYTe was also visualized with UCL imaging, demonstrating the great potential utility of the chiral nanostructure for cellular biological applications.

[1] State Key Laboratory of Food Science and Technology, Jiangnan University, 214122 Wuxi, Jiangsu, China. [2] International Joint Research Laboratory for Biointerface and Biodetection, School of Food Science and Technology, Jiangnan University, Wuxi, JiangSu 214122, China. [3] Shanghai Synchrotron Radiation Facility, Shanghai Institute of Applied Physics, Chinese Academy of Sciences, Shanghai 201204, China. These authors contributed equally: Maozhong Sun, Tiantian Hao, Xiaoyun Li, Aihua Qu. Correspondence and requests for materials should be addressed to H.K. (email: kuangh@jiangnan.edu.cn)

The study of the chiroptical activity of plasmonic nanomaterials has provoked extensive interest because their shape- and material-composition-dependent characteristics facilitate their broad potential application[1–3]. Among these nanomaterials, a growing number of DNA-based nanoassemblies not only provide a practicable route by which to fabricate possible configurations of nanomaterials in controllable ways, but also an opportunity to produce photoelectrical properties through the integrated behavior of their individual building blocks[4–7]. Significant efforts have been devoted to exploiting novel chiral materials in the fields of photonics, catalysis, electronics, analytics, and so on[8–12]. Chiral assemblies have recently become a new type of biosensor for probing intracellular molecules[13,14]. Moreover, the dependence on circular dichroism (CD) spectra could potentially allow the differentiation of the extracellular and intracellular localization of plasmonic assemblies[15]. However, the great challenge in this field is our limited knowledge of the physiological interactions of chiral assemblies with cellular metabolic processes within living organisms.

Autophagy is a basic metabolic process in which eukaryotic cells break down superfluous or dysfunctional cellular components through a lysosome-dependent pathway and recycle their biogenic constituents[16–18]. Accumulating evidence has shown that the abnormal regulation of autophagy is directly involved in many types of pathologies, including aging, neurodegeneration, cancer, and diabetes[19,20]. Therefore, the precise modulation of autophagy plays a pivotal role in regulating and maintaining normal physiological functions[21]. The activation of autophagy in living cells is probably generally induced by cellular starvation, cytokines, and even antibiotic stimuli[22]. Advanced nanomaterials for regulating cellular processes have recently received tremendous attention[23–25], and many nanoscale inducers of autophagy of various sizes, morphologies, and chemistries have been developed[26–28]. Despite the extensive efforts in this direction, there has been no research into the effects of chiral plasmonic assemblies on the control of autophagy. The obstacle in this regard is the lack of a compact unique system to accommodate combinations of imaging probes for metabolic activities that specifically respond to triggers of autophagy. This could rapidly and accurately monitor the autophagic state in living cells in real time.

The primary currency for energy in almost all cellular activities is adenosine triphosphate (ATP)[29–33], which is also used as an endogenic indicator of cell viability, cell injury, and activities regulator in many cellular processes[29–37]. Therefore, developing a nanodevice capable of responding to diverse targets with versatile signal changes is becoming the focus of much research[38,39]. The main factors determining the behavior of these devices in various applications are the geometrical configurations and surface properties. Nanoassemblies with tetrahedral shapes and topologies have shown superior plasmonic chiroptical properties in the visible range[40,41]. The continued focus of our group has been on multiplexing sensing capabilities, imaging, and therapeutic agents.

Now, in this study, we use upconversion nanoparticles (UCNPs) and yolk–shell nanoparticles (YSNPs) as the building blocks to generate a UCNP-centered YSNP tetrahedron structure (UYTe) using DNA hybridization. As illustrated in Fig. 1, YSNPs dimer is formed by DNA self-assembly. Meanwhile, one of YSNPs is modified with responsive linker peptide, FGFT (sequence: Cys-Phe-Gly-Phe-Thr), which could be hydrolyzed by the autophagic biomarker of ATG4B. Then, to obtain trimers, ATP aptamer sequence-modified UCNP is hybridized with the other YSNP dimer. Finally, the dimer and trimer are combined into a UYTe structure by DNA complementary. The prepared assembly could be optically activated in two ways, displaying a strong plasmonic CD signal and a quenchable upconversion luminescence (UCL) signal. When it encounters ATG4B, the specific cleavage of the FGFT peptide cause the disassembly of YSNP in one corner and a reduction in the CD signal, whereas the UCL intensity is restored by the activation of ATP production during autophagy. With this de novo design, the chirality of the nanodevice is further tailored by decoration with chiral D-/L-glutathione (GSH), and this nanodevice could be used as an intracellular autophagy inducer. After incubation with tumor cells, the UYTe generates a chirality-dependent autophagy-inducing activity. With the enhanced level of autophagy, the YSNPs modified with the responsive peptide are disassembled, which reduce the intracellular CD signal. The production of ATP is enhanced with the induction of autophagy, which triggers an increase in the intracellular UCL intensity in living cells. More importantly, autophagy is induced in vivo by the UYTe assembly and the corresponding production of ATP is simultaneously quantified with UCL imaging.

## Results

**Structural design and assembly of chiral UYTe.** Interior-nanogap-inlayed YSNPs (24.3 ± 2.1 nm represents mean ± standard deviation) and maleimide-modified UCNPs (19.3 ± 1.3 nm) were prepared (Supplementary Fig. 1 and Note 1)[42,43]. YSNP dimers were first fabricated through partial DNA (DNA 1 and DNA 2) complementary. An autophagy-responsive peptide that terminated in cysteine was tightly coupled to the one of the YSNPs through an Au–S bond and then amino-modified DNA 1 were connected to the carboxy group of the peptide. After hybridized with the other YSNP (sulfhydryl-terminated DNA 2 modified), a high dimer yield of 89.3% ± 1.2% were obtained based on transmission electron microscopy (TEM) images (Fig. 2a). Besides, to constructed trimer structure, YSNPs with no peptide modification were prepared and coupled directly to sulfhydryl-terminated DNAs (DNA 3-1, DNA 3-3, and DNA 4; see Supplementary Table 1), respectively. The obtained two YSNPs were then hybridized with the ATP aptamer (ATP apt)-sequence-modified UCNP. To immobilize the UCNP in the center of the tetrahedron and facilitate ATP targeting, three aptamers on the UCNP were hybridized with the tetrahedron DNA backbone, in a stem–loop structure, from which the UCNP could be released with ATP-triggered DNA disassembly (for details, see Supplementary Method). TEM images showed that the trimeric structure of two YSNPs and one UCNP was successfully generated, with a high yield of 88.1% ± 1.4% (Fig. 2b, e). Finally, UCNP-centered YSNP tetrahedron (UYTe) was formed by hybridizing the prepared assemblies with additional DNAs (DNA 3-2; see Supplementary Table 1) (Fig. 2c).

The detailed progress of the UYTe assembly process over time was recorded in TEM images. Both dimers and trimers were observed at the beginning of the assembly process (Supplementary Fig. 2). When the incubation time was extended to 3 h, the assembled tetrahedron appeared with UCNP at its center. After incubation for 9 h, the assembly was completed and confirmed with TEM images, with a yield of 84.6% ± 1.7%. One of the three ATP aptamers on UCNP was hybridized with another tetrahedron DNA backbone, which further stabilized the intercalated UCNPs. The UYTe nanostructures were composed of elemental Au and Ag (YSNP) in the apex region of the tetrahedron, and elemental Gd (UCNP) in the satellite region, which was confirmed with energy-dispersive X-ray imaging (Fig. 2d). To verify the successful assembly, the assembly process was analyzed with agarose gel electrophoresis. As illustrated in Fig. 2f, the monodispersed Au NP or YSNP (lane 1 or lane 2) migrated a greater distance than the YSNP dimer (lane 3) because of the

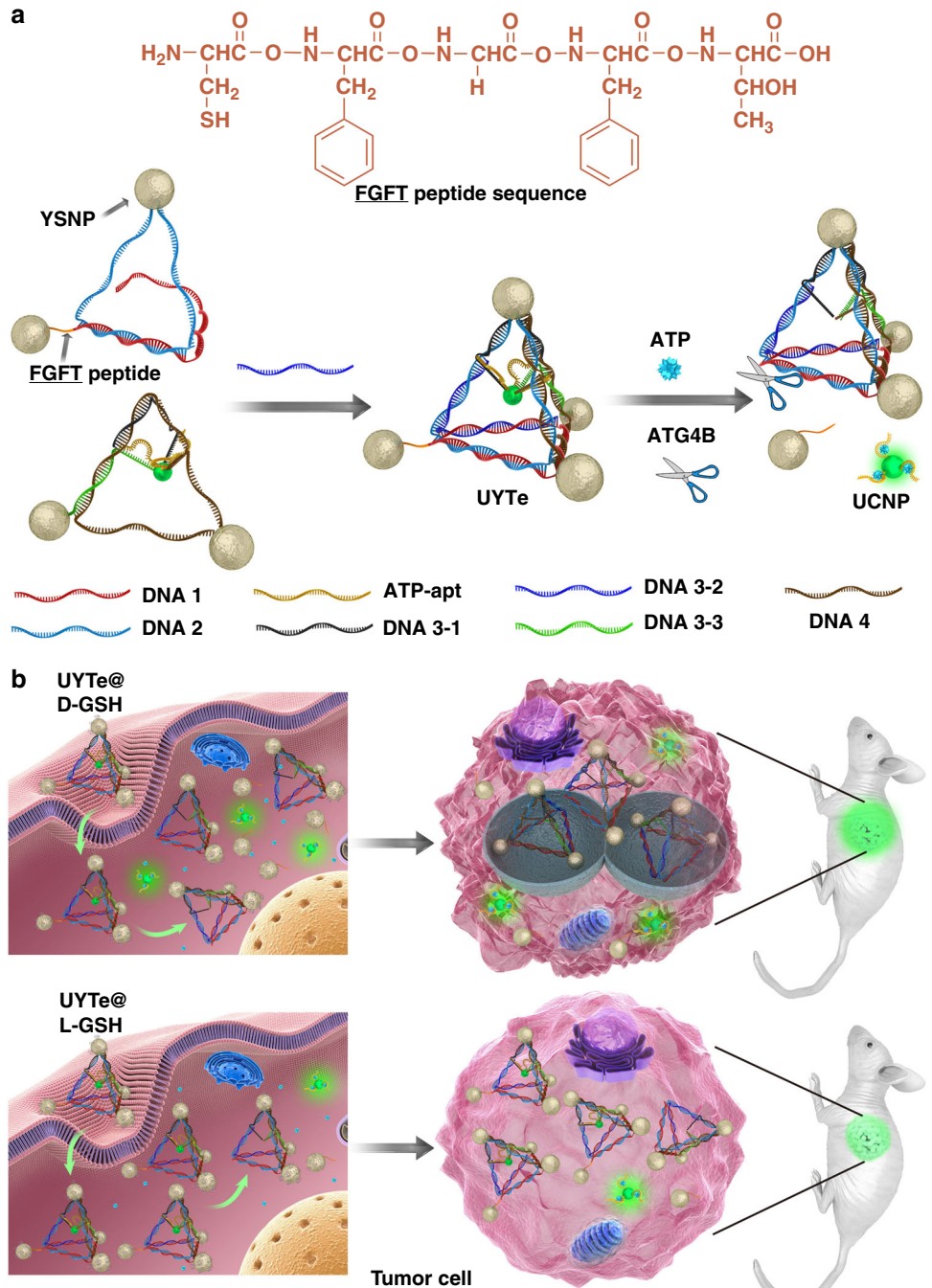

**Fig. 1** Schematic of chiral nanodevice for autophagy induction and observation. **a** The self-assembly process of UYTe and the detection principle for autophagy and ATP. **b** The induced autophagy and corresponding ATP quantitative detection both in living cell and in mice

increase in size after assembly. Notably, there was no distinct color change after further assembly with UCNPs and the migration distance was shorter after trimer formation (lane 4). The distinct band in lane 5 was the UYTe nanostructure, which migrated more slowly than the other assemblies or nanoparticles. The apparently slowest band showed a high yield, which reflected the presence of dimeric and trimeric structures. To determine the spatial arrangement of the structure, transmission electron cryomicroscopy (cryo-TEM) and concurrent 3D tomographic reconstructions strongly confirmed the chiral configuration and showed that the UCNP was encapsulated within the center of the YSNP tetrahedral assembly (Fig. 2g). The small-angle X-ray scattering (SAXS) spectra before and after the encapsulation of

UCNP further confirmed that the maximum intraparticle distances were nearly the same (Fig. 2h, Supplementary Table 2 and Supplementary Fig. 3). Dynamic light scattering was also used to characterize the assembly process, and showed a well-dispersed assembly with a hydrodynamic diameter that changed from $21 \pm 3$ nm to $29 \pm 5$ nm to $67 \pm 7$ nm (Supplementary Fig. 4).

**Regulating the optical activity of the UYTe assembly.** The changes in the optical properties of the structure during the assembly process were also measured. In the CD spectra, both the YSNP dimer and YSNP-YSNP-UCNP trimer had weak signals of $4 \pm 1.5$ and $7 \pm 1.2$ mdeg, respectively, in the region of

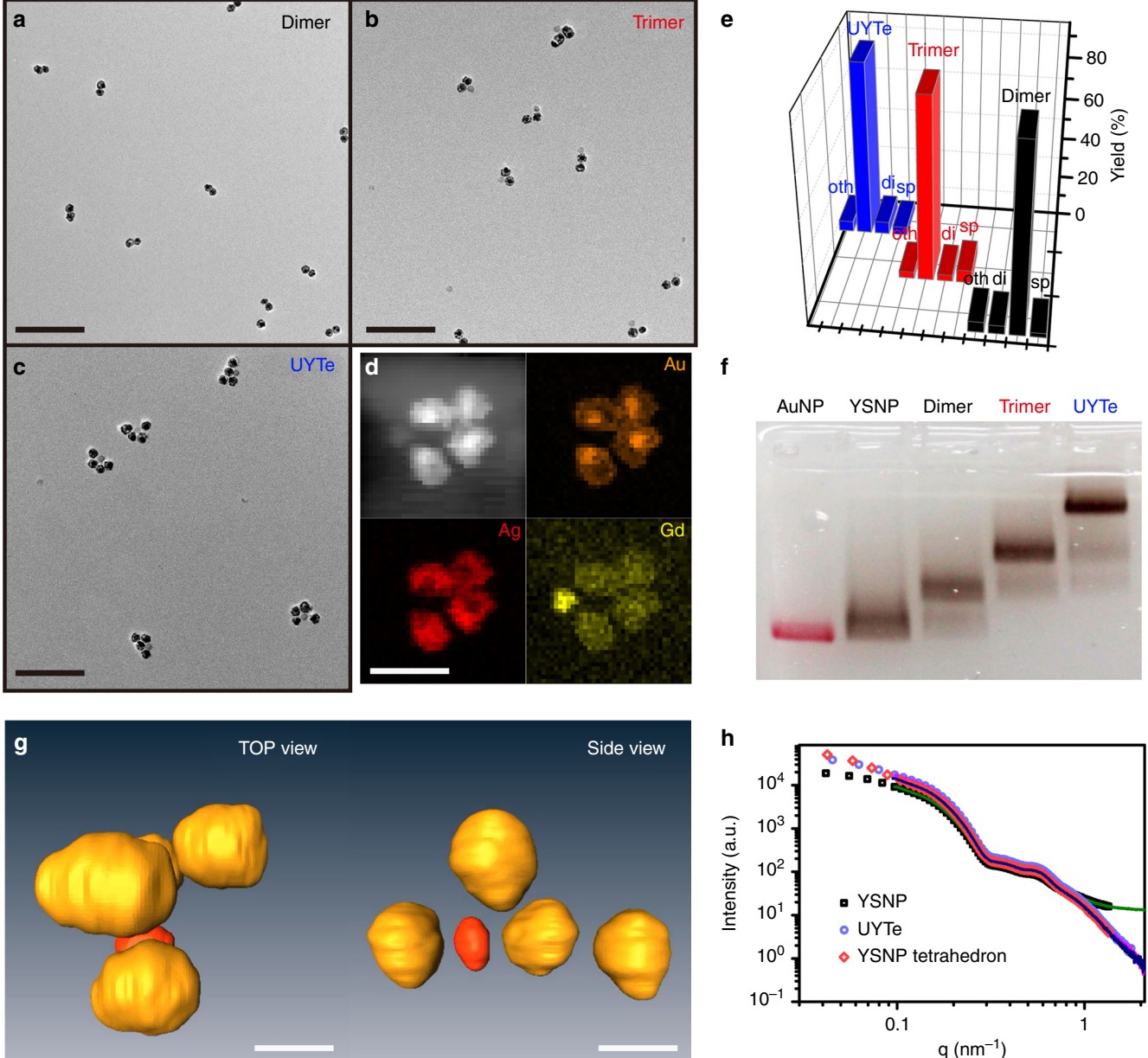

**Fig. 2** Construction of self-assembled chiral nanodevice. The TEM images of **a** YSNP dimer **b** YSNP-YSNP-UCNP trimer, **c** UYTe, scale bar 200 nm. **d** The corresponding EDX mapping images of UYTe, scale bar 50 nm. **e** The yield of YSNP dimer, YSNP-YSNP-UCNP trimer, and UYTe during the self-assembly progress. **f** The electrophoresis image of AuNP, YSNPs, YSNP dimer, YSNP-YSNP-UCNP trimer, and UYTe. **g** The corresponding 3D reconstruction cryo-TEM tomography image of UYTe, scale bar 20 nm. **h** The SAXS spectrum of YSNPs, UYTe, and YSNP tetrahedrons. "sp" means single-particle, "di" and "tri" stand for assemblies consist of two and three-particle, respectively, "oth" means other assemblies. All experiments were performed in triplicate

470–596 nm (Fig. 3a). When the dimer and trimer were mixed, the change in the CD signal recorded their hybridization. As the incubation time increased, the CD signal intensity increased. After the assembly reaction had proceeded for 9 h, the UYTe nanostructure showed a strong CD intensity of about 26 ± 1.3 mdeg at 511 nm of the plasmonic peak (Supplementary Fig. 5). It is noteworthy that there was no obvious signal for each building block, indicating that the CD signal originated from the chiral conformation of the assembly[44].

The variation in UCL during the assembly process was inspected during excitation at 980 nm. Three distinct UCL peaks for UCNP were observed at 528, 541, and 655 nm, which is consistent with previous reports (Fig. 3b)[43]. When the UCNP was assembled with the YSNP dimer, the quenched UCL intensity

observed was ascribed to the close proximity of the plasmonic YSNP and UCNP. After further assembly, the UCL of the UYTe nanostructures was clearly quenched, and showed only a weak signal.

The chiroplasmonic activity of the UYTe nanostructures was then tailored by decorating the surface of the YSNPs with chiral molecules of D-/ L-GSH. As a comparison, a YSNP tetrahedron without a UCNP center was also prepared. The optimum concentration of GSH coupled to the YSNPs was determined from the CD spectra (Supplementary Fig. 6), which showed that the CD signal of YSNP increased with the further addition of L-GSH. The CD intensity was highest when the L-GSH concentration reached 5 μM, after which the signal decreased with further increases in L-GSH because of the irreversible aggregation.

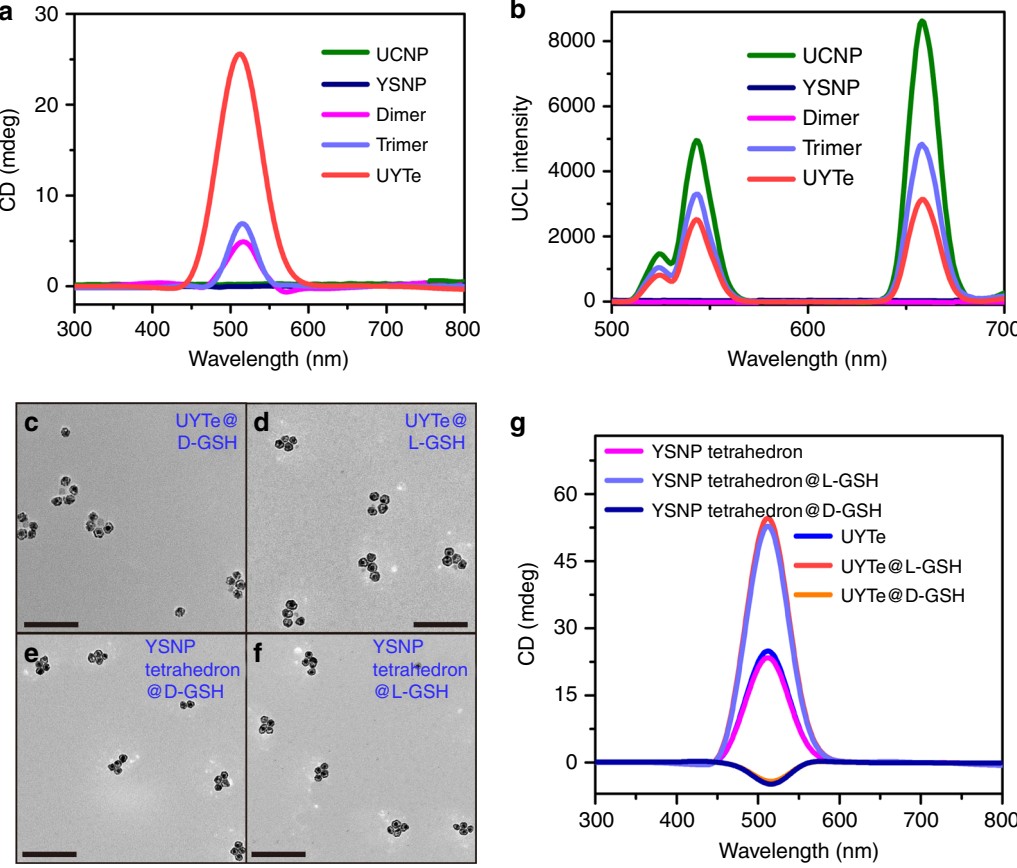

**Fig. 3** Spectroscopic characterization of self-assembled chiral nanodevice. The **a** CD and **b** UCL spectrum of UCNP, YSNPs, YSNP dimer, YSNP-YSNP-UCNP trimer, and UYTe assemblies. The TEM images of **c** D-GSH and **d** L-GSH-modified UYTe. The TEM images of **e** D-GSH and **f** L-GSH modified YSNP tetrahedrons, scale bar 200 nm. **g** The CD spectrum of D-/ L-GSH modified UYTe and YSNP tetrahedrons. All experiments were performed in triplicate

Therefore, 5 µM L-GSH was chosen for subsequent assembly reactions. From TEM images, both the D- and L-GSH-modified UYTe nanodevice were well dispersed (Fig. 3c, d). The YSNP tetrahedron structure was also determined after modification with D- or L-GSH, which showed a high yield of about 85.7% ± 1.2% (Fig. 3e, f). The CD signals of both kinds of assembly displayed nearly the same intensity (Fig. 3g). After L-GSH modification, the CD intensity of the UYTe nanostructure was 54.7 ± 1.5 mdeg, a little larger than that of the YSNP tetrahedron assembly (52.5 ± 2.1 mdeg). However, if the assembly was decorated with D-GSH, the UYTe and YSNP tetrahedron assemblies showed negative CD signals in the same region (−3.97 ± 1.1 and −4.52 ± 1.4 mdeg, respectively). The variation in the CD signal can be explained as follows. First, the YSNP tetrahedron assembly displayed a positive CD signal arising from its chiral tetrahedron shape. Second, with GSH modification, the YSNP shell structure generated an intensive plasmon-induced chiral signal from the chiral GSH, which is consistent with previous reports[45,46]. Therefore, the chirality of the UYTe was mainly attributable to two factors, its chiral configuration and plasmonic-induced chirality. Because the CD signal of L-GSH modified YSNP was positive, the CD intensity of UYTe was increased after L-GSH coating. On the contrary, the CD signal of YSNP modified with D-GSH was negative and the UYTe showed a weak negative signal because the counteracted signal between chiral structure and the induced chirality. Notably, the induced chirality was much stronger than the structural chirality, which caused the reversal of the chirality. Third, without UCNP embedded in the nanostructure, the DNA backbone was unchanged. The YSNP tetrahedron nanostructure

had nearly the same CD signal as the UYTe structure, indicating that the CD signal mainly originated from the tetrahedron.

**Chiral UYTe nanostructures for autophagy induction dynamic monitoring.** After the assembly was characterized, we tested the feasibility of autophagy and ATP monitoring in solution with the UYTe nanostructure without GSH modification. The CD signal at 511 nm decreased as the ATG4B concentration (in PBS) increased, whereas the UCL intensity remained almost unchanged (Supplementary Fig. 7)[27]. This was attributed to the disassembly of the UYTe structure by the enzymatic digestion of the autophagy-related peptide FGFT by ATG4B, and the signal of the UCNP embedded in the tetrahedron was quenched by the three remaining YSNPs (Supplementary Fig. 8). Interestingly, there was no obvious changes in the CD spectrum in the plasmonic region when only ATP was added. The greater the ATP content, the higher the intensity of UCL on the UCL spectrum (Supplementary Fig. 9), which was attributed to the dissociation of UCNP from the UYTe structure, as confirmed in TEM images (Supplementary Fig. 10). These data demonstrate that the UYTe assembly can be used to monitor autophagy and ATP content.

As is know that autophagy-related response relies on the expression of ATP. It makes sense to dynamic monitoring the ATP variation during autophagy induction[47–50]. The UYTe assembly with no GSH modification was then incubated with human breast cancer cells (MCF-7 cells, ATCC® HTB-22™) (Supplementary Figs. 11, 12). To test the quantitative detection of intracellular ATP, the cells were treated with an ATP inhibitor or

inducer, respectively. The cells were then treated for 1 h with 20 μg/mL of oligomycin (ATP inhibitor) or 10 μg/mL of oligomycin, 0.1 M PBS at pH 7.4, 50 μg/mL of etoposide (ATP inducer), or 100 μg/mL of etoposide. The cells incubated with high-concentration oligomycin produced only a limited UCL signal on confocal microscopic images, indicating the successful inhibition of ATP activity. However, when treated with the ATP inducer, the cells displayed distinctly enhanced UCL intensity compared with that in the PBS-treated group. This clearly demonstrates that the UCL signal of the UYTe assembly recovered as the intracellular ATP concentration increased (Fig. 4a). This result was confirmed with a commercial ATP ELISA kit, which showed that the ATP concentrations in groups were 1.1, 1.8, 2.3, 2.9, and 4.3 mM, respectively (Supplementary Fig. 13).

To test whether the chiral plasmonic nanostructures exerted an autophagy-inducing effect in living cells, UYTe assemblies modified with different GSH enantiomers (40 nM) were incubated with MCF-7 cells. A standard autophagy detection kit was used to study the degree of autophagy in the cells. As evident in confocal microscopy images, the cells treated with the D-GSH-coated UYTe structure produced a green signal (reflecting the intensity of autophagy) that was stronger than the signal induced by treatment with the clinical autophagy inducer, rapamycin, indicating that the chiral plasmonic nanoassembly effectively activated autophagy in living cells. In contrast, the L-GSH-coated UYTe assembly and the unmodified UYTe assembly produced only limited signals (Fig. 4b). To figure out the mechanism of chirality-dependent autophagy activity, the Au content of chiral GSH modified UYTe during the incubation was studied from ICP-MS and bio-TEM. It showed that the extensive and dense aggregation of D-GSH modified UYTe in living cell was observed. While the L-GSH modified UYTe only had limited aggregation in endocytosis vesicles (Supplementary Fig. 14). Also, from ICP-MS, the Au content in living cells after incubated with D-GSH UYTe had the higher residual content when the incubation time was 12 h (Supplementary Fig. 15). Moreover, the accumulated D-GSH-modified UYTe further enhanced the intracellular oxidative stress which was measured by H2DCFDA (Invitrogen™) (Supplementary Fig. 16). The ATP concentration was also monitored during this autophagy activity. Interestingly, the intensity of the UCL signal increased as the degree of autophagy increased. Both kinds of signal were highest during incubation with identical D-GSH-coated UYTe assemblies (Supplementary Fig. 17), demonstrating that the ATP concentration increased as the degree of autophagy increased in living cells (Supplementary Fig. 18). This was mainly attributable to enzyme-associated autophagy, the activation of which induced the corresponding production of ATP to fuel the degradation of the cellular components[47]. Notably, the ATP concentration in the living cells ($5 \times 10^7$) after incubation with the D-GSH-coated UYTe assembly (40 nM) was 3.76 mM, which was calculated from the standard curve shown in Supplementary Fig. 13.

Further evidence of the successful induction of autophagy was provided by bio-TEM images (Fig. 4c and Supplementary Fig. 19-23), which showed autophagosomes and autolysosomes in the cytoplasm of both the D-GSH-coated-UYTe-treated and rapamycin-treated groups during autophagy. However, there was no obvious difference between the cells treated with the L-GSH-coated UYTe assembly and the PBS-treated cells, which was consistent with confocal microscopic images. Western blotting was also performed to check the accumulation of the microtubule associated protein light chain 3 II (LC3-II) (Fig. 4d).

The intracellular UCL and CD signals were also recorded after the incubation with chiral UYTe. As shown in Fig. 4e, f, the UCL signal in the cells treated with the D-GSH-coated UYTe assembly

was highest (7947 ± 23 at 655 nm), whereas the absolute value of CD intensity (14.4 ± 1.6 mdeg) was enhanced as compared with the extracellular signal (3.97 ± 1.1 mdeg). It was because that the chiral UYTe conformation was disassembled under autophagy, indicating the high degree of autophagy induced by D-GSH-coated UYTe. Also, the group with Rapa treatment showed increased UCL (6281 ± 18 at 655 nm) due to the induced autophagy while the CD intensity was decreased (8.5 ± 1.3 mdeg) as compared with the corresponding extracellular intensity (26 ± 1.3 mdeg).

However, the cells treated with the L-GSH-coated UYTe assembly displayed low UCL intensity and an unchanged CD signal as in solution. These results confirmed that as the degree of autophagy in living cells increased, the ATP concentration increased, causing the disassembly of the UYTe structure (Supplementary Fig. 24).

Based on this finding, the autophagy degree of D-GSH coated UYTe assembly with different concentration was investigated in detail. From Supplementary Figs. 25, 26, the MCF-7 cells in same amount were incubated with different concentration of D-GSH modified UYTe. Both two signals were enhanced as the increased concentration of assembly, demonstrating that the autophagy degree was heavily dependent on the assembly content in cells (Supplementary Figs. 27–29). Different cell type was also applied for the chiral plasmonic autophagy inducer. Human cervical cancer cell (HeLa, ATCC® CCL-2™) and mouse breast carcinoma cell (4T1, ATCC® CRL-2539™) were incubated with D-GSH modified UYTe for 12 h and the enhanced autophagy degree and ATP generation which proved the general applicability of our assembly (Supplementary Fig. 30). Detailed autophagy-inducing of D-GSH modified UYTe during the incubation were studied, which showed that the assembly began to induce autophagy when they entered the cell after 30 min (Supplementary Fig. 31).

These results indicate that treatment with the D-GSH-coated UYTe effectively activated autophagy within 12 h. To be noticed, it was found that after the YSNPs that had assembled into tetrahedrons were decorated with chiral molecules, the efficiency of autophagy induction was greatly enhanced which was two times of the previous reported[28,51], demonstrating that the successfully established chirality-specific plasmonic nanoassembly accelerated the induction of autophagy. This can be explained as follows. First, the increased size of the particles after assembly may increase the accumulation of autophagosomes[51]. Second, D-GSH is an unnatural amino acid, which cannot be used in cellular metabolism[22]. Consequently, the undigested D-GSH-modified UYTe accumulated in the cytoplasm, leading to autophagy (Supplementary Figs. 32–35). Third, the accumulated D-GSH-modified UYTe further enhanced the oxidative stress in living cells (Supplementary Figs. 36–38)[52–54], which would have increased the intracellular content of the D-GSH-coated assembly and accelerated autophagy.

**In vivo autophagy induction and imaging.** To evaluate the effects of the chiral UYTe assembly in vivo, a nude mouse model was prepared with xenografted tumors treated by the inoculation of MCF-7 cancer cells into their right flanks. The GSH-modified UYTe assembly (2 mg/mL in 200 μL of PBS) was injected intratumorally and the UYTe assembly with no GSH coating was administered to the mice with PBS or rapamycin solution. Under 980 nm irradiation, no obvious UCL signal was observed before injection (Fig. 5a). After incubation for 12 h, a marked UCL signal from the disassembly of the UYTe structure was observed at the tumor sites in all the groups, confirming that the UYTe assembly can sense ATP in living mice. Interestingly, the mice treated with D-GSH-modified UYTe or rapamycin showed enhanced UCL signals resulting from the autophagy-associated generation of

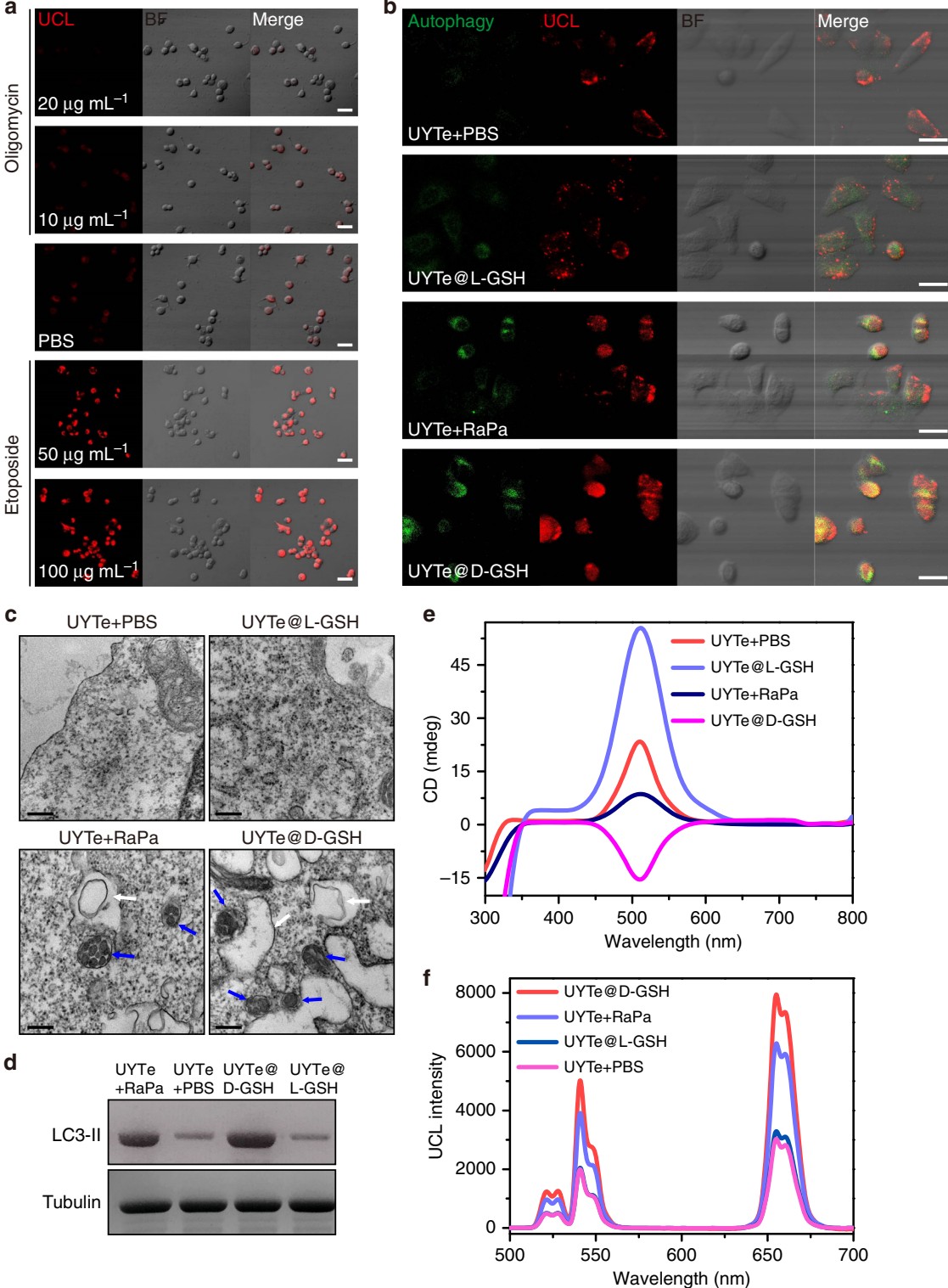

**Fig. 4** Chirality-dependent autophagy inducing and ATP quantification in living cells. **a** The UCL confocal images of MCF-7 cells ($5 \times 10^7$) treated with 20 µg mL$^{-1}$, 10 µg mL$^{-1}$ oligomycin (ATP inhibitor), PBS, or treated with 50 µg mL$^{-1}$, 100 µg mL$^{-1}$ etoposide (ATP inducer) after incubated with UYTe (40 nM), scale bar 20 µm. **b** The confocal images, scale bar 20 µm, and **c** bio-TEM images of MCF-7 cells ($5 \times 10^7$) treated by UYTe (40 nM) with PBS, L-GSH modified UYTe (40 nM), UYTe (40 nM) with Rapamycin (autophagy inducer 10 µM), and D-GSH modified UYTe (40 nM) for 12 h (GSH concentration is 5 µM), scale bar 200 nm. **d** The corresponding western blot assay of MCF-7 cells in **b**. The blue arrow indicates the autophagosomes and the white arrow means the autolysosomes. The intracellular **e** CD intensity and **f** UCL intensity of MCF-7 cells after treatment. All experiments were performed in triplicate

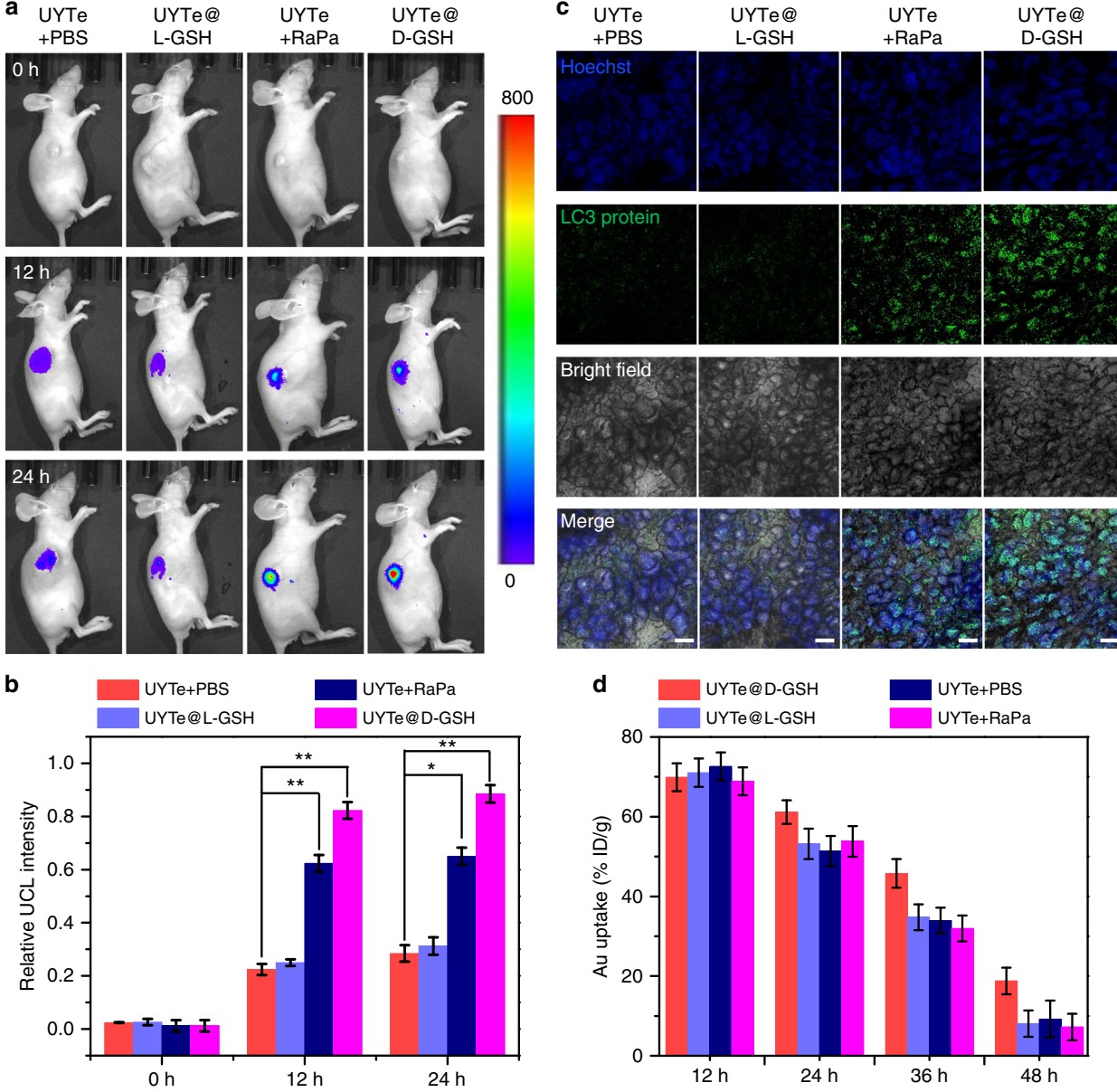

**Fig. 5** In vivo autophagy effect induced by the chiral UYTe nanodevice. **a** The in vivo UCL imaging (800 mean the maximum value, 0 mean the minimum value), **b** the UCL intensity statistics, and **c** the fluorescence images for cryosections before and after injection of PBS, UYTe (40 nM) in PBS, L-GSH modified UYTe (40 nM), UYTe (40 nM) with Rapamycin (autophagy inducer 10 μM) and D-GSH modified UYTe (40 nM) after 24 h, scale bar 200 μm. **d** The intratumor bio-distribution of Au amounts after 48 h treatment was measured by ICP-MS. The data are shown as the mean ± s.d. ($n = 3$). *$p < 0.05$, **$p < 0.01$ (Student's $t$ test)

ATP. When the incubation time was extended to 24 h, a marked change in the UCL signal in the D-GSH-modified UYTe group was observed, indicating that autophagy was efficiently induced by the chiral plasmonic nanoassembly in living mice (Fig. 5b). After treatment for 24 h, tumor cryosections were examined with immunohistochemical staining. The autophagy-related LC3 protein was stained with an antibody, and appeared as green fluorescence in the tissues[55]. Consistent with the UCL intensity, the D-GSH-modified UYTe group showed the highest LC3 expression, whereas the group treated with the same amount of L-GSH-modified UYTe showed only limited fluorescence in the tissues (Fig. 5c). The Au content in the tumors at 24 h post-injection was measured with inductively coupled plasma mass spectrometry. The D-GSH-modified UYTe showed clearly delayed excretion of the NPs, which may explain the substantial degree of autophagy

they induced (Fig. 5d). These results confirm the strong autophagy-inducing ability of the D-GSH-modified UYTe structure both in solution and in vivo, which is mainly attributable to its intracellular accumulation.

## Discussion

In conclusion, a chiral UYTe nanodevice successfully prepared with DNA-based self-assembly. The UYTe structure produced tunable CD signals that originated from both the plasmon-induced chirality and the chiral conformation, as confirmed with 3D tomography. These findings demonstrate that this D-GSH-modified UYTe nanostructure induces much greater autophagy than the L-GSH-modified assembly. Furthermore, autophagy was successfully induced in tumor-bearing mice treated with the

D-GSH-modified UYTe, and the process could be monitored with ATP-dependent UCL imaging. This ingenious nanodevice not only provides a guide for the fabrication of chiral platforms, but also has great potential utility in discerning the interactions between chiral inorganic materials and cellular metabolic activities in living systems.

## Methods

**Self-assembly of UYTe nanostructure.** UCNP with maleimide modification was diluted 100-fold, and then functionalized with thiolated single-stranded DNA (ssDNA, synthesized from Sangon Biotech (Shanghai) Co., Ltd.) by mixing in a ratio of 1:5 and incubating them for 10 h in 10 mM Tris-HCl (pH 7.5) containing 50 mM NaCl. The resulting nanoparticle–DNA conjugate was obtained with ultrafiltration at 8000 × g for 10 min and resuspended in Tris-HCl.

YSNPs, which prepared beforehand, were functionalized with thiolated ssDNA (DNA 3-3 and 4) by mixing with the ssDNA in a ratio of 1:5 and incubated for 10 h in 10 mM Tris-HCl (pH 7.5) containing 50 mM NaCl. The samples were then centrifuged at 10,000 × g for 10 min to remove any uncoupled oligonucleotides from the solution. The supernatant was removed and the pellet was resuspended in Tris-HCl.

YSNPs were modified with a peptide whose terminal ligand was a carboxyl group, and functionalized with ssDNA (DNA1 and DNA2) by mixing the shell with 0.5 mM 1-(3-dimethyl-aminopropyl)-3-ethylcarbodiimide hydrochloride and 0.05 mM N-hydroxysuccinimide for 6 h to activate the carboxyl group. The ssDNA was added to the shell in a ratio of 1:5 and incubated for 10 h in 10 mM Tris-HCl (pH 7.5) containing 50 mM NaCl. The samples were then centrifuged at 10,000 × g for 10 min to remove any uncoupled oligonucleotides from the solution. The supernatant was removed and the pellet was resuspended in Tris-HCl.

YSNP dimer was prepared by mixing two YSNPs, each of them modified with complementary DNA conjugates (DNA1 and DNA2) at 90 °C for 5 min and cooled slowly to room temperature, respectively. The samples were then centrifuged at 5500 × g for 10 min to remove any uncoupled single particles from the solution, and the pellet was resuspended in Tris-HCl.

YSNP-YSNP-UCNP trimer was prepared by adding shell NPs modified with DNA3-3 or DNA4 to UCNP modified with DNA-Apt, after which DNA3-1 was mixed with them. The DNA mixture was heated for 5 min at 90 °C and then cooled to room temperature. Finally, the sample was centrifuged at 5000 × g for 10 min to remove any uncoupled single particles, and the pellets were resuspended in Tris-HCl.

YSNP dimer and YSNP-YSNP-UCNP trimer were mixed with DNA3-2 for 8 h to complete the assembly of the framework, and then centrifuged at 4000 × g for 10 min to isolate UYTe.

**Frozen-hydrated sample preparation.** Colloidal gold beads (10 nm) were added to the sample as fiducial markers and the aqueous sample was applied to an electron microscope gird. The grid was dried to near-complete dryness and then plunged into liquid ethane for the rapid vitrification of the sample, which was then stored in liquid nitrogen until use.

**Cryo-TEM imaging.** The cryo-TEM images were obtained with a Tecnai F20 transmission electron microscope (Tecnai F20, FEI) equipped with an Eagle 4 K × 4 K multiport CCD camera (FEI) under a 200-kV electron accelerating voltage. The tilt series (single-axis tilt) was collected from −50° to + 66° at intervals of 2° (the defocus value set at −3 μm, under ×11,500 magnification) using the Serial EM software.

**3D reconstruction and rendering.** The tilt series was aligned and reconstructed with IMOD, according to the software manual, to convert the tomograms into a 3D structural model. Reconstruction was performed with a simultaneous iterative reconstruction technique with five iterations. These structural models were then manually segmented and visualized with the Chimera program.

**CD and UV-vis measurement.** MCF-7 cells were seeded in a six-well plate with a density of $10^5$ cells per well. The UYTe (40 nM) with PBS, L-GSH modified UYTe (40 nM), UYTe (40 nM) with Rapamycin (autophagy inducer 10 μM), and D-GSH modified UYTe (40 nM) were co-cultivated with MCF-7 cells for 12 h at 37 °C, respectively. The cells were collected using standard trypsin-based lift-off protocol and washed with PBS three times. Then, the cells were re-dispersed in 200 μL PBS and the chiroptical activity and UV-vis absorption was characterized by Chirascan (Applied Photophysics Ltd). The optical path length was 1 cm.

**Small-angle X-ray scattering data analysis.** The P(r) (pair-distance distribution (PDDF)) function, which represents for particles the number of distances within the particle. i.e., the number of lines with lengths $r$ that are found in the combination of any small volume element $i$ with any other volume element $k$. For spherically symmetric particles, the P(r) is defined as Equation 1:

$$P(r) = \frac{1}{2\pi^2} \int_{q=0}^{\infty} q^2 I(q) \frac{\sin qr}{qr} dq \qquad (1)$$

The diameter of particles and the maximum intraparticle distance ($D_{max}$) were obtained from the P(r) function. P(r) function, the P(r)-derived radii and the maximum intraparticle distance ($D_{max}$) were obtained using GNOM. The $D_{max}$, since P(r) drops to zero at $r = D$.

**Cell lines and incubation conditions.** MCF-7, HeLa, and 4T1 cells were purchased from the American Type Culture Collection (Manassas, VA, USA). The cells were cultured in Dulbecco's modified Eagle's medium containing 10% fetal bovine serum and 1% penicillin/streptomycin at 37 °C under a 5% $CO_2$ atmosphere.

**Animal tumor models.** All animal studies were performed according to institutional ethical guidelines and were approved by the Committee on Animal Welfare of Jiangnan University. Five-week-old female nude mice (average weight 15.5 g) were subcutaneously inoculated above the right flanks with $5 \times 10^6$ MCF cells suspended in 50 μL of cold PBS. The in vivo experiments were performed after 3 weeks.

**In vivo imaging of ATP.** UCNP luminescent imaging was performed with the IVIS Lumina II in vivo imaging system (Caliper Life Sciences, Inc.) at various time points (0, 12, or 24 h). The D-GSH-modified UYTe assembly (200 μL; amount of Au, 2 mg/mL) was intravenously injected into the nude mice. After 0, 12, and 24 h, fluorescent images of the flanks of the live mice were taken under excitation with a 980 nm laser.

**Life sciences reporting summary.** Further information on experimental design and reagents is available in the Life Sciences Reporting Summary.

## Data availability
The data supporting the findings of this study are available within the paper and its Supplementary Information files and are available from the corresponding author upon reasonable request.

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

## Acknowledgements

This work is financially supported by the National Natural Science Foundation of China (21631005, 21673104, 21522102 and 21503095). We thank Dr. XR Miao for help with the small-angle X-ray scattering (SAXS) measurements. This research was performed at the SAXS beamline instrument at BL16B1 at the Shanghai Synchrotron Radiation Facility and sponsored by Shanghai Sailing Program (17YF1423800).

## Author contributions

H.K. designed the experiments. M.S. conducted autophagy inducing, spectroscopic measurements, and data analysis. T.H. assembled the nanodevice. X.L. contributed to the S-SAXS characterization. A.Q. carried out the cell and in vivo experiments. L.X. and C.H. took Cryo-TEM and 3D reconstruction. H.K. carried out the histopathological examination. H.K. and C.X. supervised the work and analyzed the results. H.K. and C.X. co-wrote the manuscript. All authors discussed the results and commented on the manuscript.

## Additional information

**Competing interests:** The authors declare no competing interests.

