## [Peer Review File · Nature Communications]

Reviewers' comments:

Reviewer #1 (Remarks to the Author):

This work reports the fabrication of UCNP-centered YSNP tetrahedron structure (UYTe) through DNA-based self-assembly. The researchers believe that the proposed structure can be used to induce and monitor the autophagy when modified with D-GSH. TEM, electrophoresis, and SAXS characterizations are employed to verify the assembly of UYTe. The researchers subsequently investigate the changes in CD spectra and UCL intensity of the structure upon assembly. Finally, the authors demonstrate that, in comparison with L-GSH-modified assembly, D-GSH-modified UYTe induces much greater autophagy in vivo and in vitro, which can be monitored with ATP-dependent UCL imaging (Figure 3 and Figure 4). While the experimental work is carefully carried out, I personally find the paper lacks novelty. There are also many technical issues with the manuscript. Therefore, I do not feel this work represents a considerable breakthrough that warrants its publication in Nature Communications.

Suggested improvements:

1. In this manuscript, my understanding is that the authors want to present two arguments: one is that the D-GSH-modified UYTe can induce autophagy; the other is the induction of autophagy can be monitored by ATP-released UCL fluorescence. However, one should bear in mind that many types of materials have been reported to induce autophagy such as gold nanoparticles (Biomaterials, 2010, 31, 5996-6003) and upconversion nanoparticles (J. Am. Chem. Soc. 2015, 137, 6550–6558). Would these materials have the same effect as D-GSH-modified UYTe upon coating with GSH?
2. Other researchers have examined the effects of QDs, coated with different chiral forms of GSH, on induction of autophagy (Angew. Chem. Int. Ed. 2011, 50, 5860–5864). The chirality-dependent activation of autophagy was. The main difference is that in the published work the L-GSH has proven more effective than D-GSH.
3. It is true that ATP can release UCL fluorescence, but I do not think the increment of ATP are only caused by autophagy. ATP provides energy for cell growth, proliferation, and even apoptosis. Thus, I do not think UCL fluorescence can be directly linked to autophagy.
4. It puzzles me that the authors use Au@Ag@Au instead of just Au nanoparticles to construct UYTe structures (Figure 1a).
5. Page 4, Line 85, "one of YSNPs was modified with peptide, ATG4B." and Page 4, Line 90, "When it encountered ATG4B" is contradiction. What is ATG4B, an enzyme or a peptide?
6. There are quite many similar abbreviations throughout the manuscript.
7. Figure 3C looks blurry. For improvement, authors can refer to Figure 2C in the published paper (Angew Chem Int Ed 2011, 50, 5860–5864)
8. What is FGFT? I cannot find the description in text.

Reviewer #2 (Remarks to the Author):

In the current manuscript, the authors come up with a novel idea of generating chiral, tetrahedral DNA programmed devices to induce and monitor Autophagy in cells and tissues. The authors explore the chiral properties of fluorescence responsive biomolecules like GSH and ATP as a reporter. They successfully couple the nanodevices with Autophagy sensing peptide (ATGB4), the recognition of which with cellular targets, triggers the disassembly of nanodevice leading to fluorescence readout corresponding to extent of Autophagy and the involvement of ATP in this cellular process.

However, the manuscript suffers from novelty and some major factors (see below) need a strong attention before the manuscript becomes suitable for publication in Nature Communications.

1. The concept is certainly good and innovative. However, the novelty of the idea is compromised since most of these things are already known in literature and published results (ref 22-28, 38-39). The main problem is that this device induces autophagy and then the authors could sense it. Is it possible to only sense the autophagy and not induce it. For example, if they take cells grown in normal FBS and the serum starved cells, which have high autophagy, and then use the same device without ATGB peptide, is it possible to still sense the autophagy. Mostly, autophagosomes are lysosomes or autophagolysosomes. And bulk of the recent literature on DNA nanodevices show that DNA devices post uptake are targeted to lysosomes. So in principle it should be possible to generate only a sensor and not the inducer.

2. DNA tetrahedron is definitely a chiroptical molecule, but not in this case. In this case, the chirality is induced only by D or L - GSH. So, the main focus should be on the chirality induced by ligand and not by DNA. This is actually evident from fig. 3B where the authors show that L and not D-GSH decorated DNA devices can induce autophagy.

3. Page 4, Please introduce for the first time what is ATGB4. Now in fig. 1 there is no mention of ATGB4. Is it FGFT in the fig 1?

4. Page 5, 10 and other places: In situ / in vitro should be replaced by in cells or living cells.

5. Fig. 1: Is there a control on number of FGFT peptides per gold nanoparticles or number of GSH? Ideally it should be one ligand, one nanoparticle but I guess for the cellular processes, multifunctionalization might be needed.

Also, would have been excellent to see if different degrees of functionalization of device behave differently in cells. But that's just a suggestion.

6. Page 6: Fig 1d is not mentioned in the fig legend later. please check.

7. Page 7: The size of the particles: Around 70 nm. Now, if we see any endocytic process like clathrin mediated, non-clathrin, caveolae,... in all the mechanisms the maximum size of the particles which gets endocytosed is not bigger than 50 nm. Thus, after 50 nm most of the particles get phagocytosed and endocytosed. The authors should do different times of incubation of these nanodevices to cells starting from few seconds, to mins to hours. Ofcourse the autophagy will begin only after 30-40 mins (as against what authors show in Supplementary fig. S17 for 12 h), but it will give clue to authors how these devices enter into the cells and when they start to induce autophagy.

8. Page 11: the authors claim that autophagy is dependent on ATP in cells. Please cite the reference. Without the reference, it is difficult to accept this claim.

9. The TEM images are very nice and self explanatory. However, one caution - are there some standards to show that the vesicles are autophagosomes or autolysosomes. Like some marker antibody coupled to gold nanoparticles used as markers for these organelles.

10. Fig. 4 and 3: I suggest that authors also use LC3 antibody in fig. 3 where they take only cells and show that indeed the levels of autophagy has increased.

Also, please zoom in some of the regions of cells to show that indeed the staining shows punctate structures of autophagosomes and not just cytosolic or non-specific cellular staining.

11. Fig.1 - the schematic of the device operation is very complicated to understand. The authors should definitely try to simplify it.

12. Technical query - how did the authors measure UV and CD in cells? I could not find the procedure in methods or supplementary information.

No issues with statistics and data analysis. All the ethics and rules for using cells lines and animal models seem to be followed by the authors.

Overall, good work but will need further revision before final decision.

Responses to Reviewers

(NCOMMS-18-15064)

Reviewers' comments:

Reviewer #1 (Remarks to the Author):

This work reports the fabrication of UCNP-centered YSNP tetrahedron structure (UYTe) through DNA-based self-assembly. The researchers believe that the proposed structure can be used to induce and monitor the autophagy when modified with D-GSH. TEM, electrophoresis, and SAXS characterizations are employed to verify the assembly of UYTe. The researchers subsequently investigate the changes in CD spectra and UCL intensity of the structure upon assembly. Finally, the authors demonstrate that, in comparison with L-GSH-modified assembly, D-GSH-modified UYTe induces much greater autophagy in vivo and in vitro, which can be monitored with ATP-dependent UCL imaging (Figure 3 and Figure 4). While the experimental work is carefully carried out, I personally find the paper lacks novelty. There are also many technical issues with the manuscript. Therefore, I do not feel this work represents a considerable breakthrough that warrants its publication in Nature Communications.

Reply: We thank the reviewer for his/her hard work and constructive comments. Chiral nanodevice of UYTe was fabricated for first time to induce autophagy. As illustrated in **Scheme 1**, the prepared assembly could be optically activated in two ways, displaying a strong plasmonic CD signal and a quenchable UCL signal. ATG4B is one of the most important autophagy-related cysteine proteases and has been utilized as a potential autophagic biomarker. FGFT (sequence: Cys-Phe-Gly-Phe-Thr) is a responsive peptide sequence that could be hydrolyzed by the autophagy-related proteases, ATG4B (*Autophagy*, 2015, 11(2): 403-415; *Autophagy*, 2011, 7(9):

1052-1062.). When UYTe encountered ATG4B, the specific cleavage of the FGFT peptide caused the disassembly of YSNP in one corner and a reduction in the CD signal, whereas the UCL intensity was restored by the activation of ATP production during autophagy. With this de novo design, the chirality of the nanodevice was further tailored by decoration with chiral D-/L- GSH, and this nanodevice could be used as not only an intracellular autophagy inducer but also an autophagy sensor ([See the response to Reviewer 2[#]](#)).

After incubation with tumor cells, the UYTe generated a chirality-dependent autophagy-inducing activity. It showed that the D-GSH modified UYTe showed higher autophagy inducing ability than L-GSH modification. With the enhanced level of autophagy, the YSNPs modified with the responsive peptide were disassembled, which reduced the intracellular CD signal. The production of ATP was enhanced with the induction of autophagy, which triggered an increase in the intracellular UCL intensity in living cells. More importantly, autophagy was induced *in vivo* by the UYTe assembly and the corresponding production of ATP was simultaneously quantified with UCL imaging.

The novelty in our manuscript are as follows:

1. A chiral nanodevice made of yolk–shell nanoparticles tetrahedron centralized with UCNPs (UYTe) has been fabricated with the yield of $84.6\% \pm 1.7\%$ for first time.
2. It has been proved the chiral UYTe nanodevice showed chirality dependent autophagy-inducing ability and ATP production in living cell. The mechanism are as follows: first, the increased size of the particles after assembly could increase the accumulation of autophagosomes. Second, D-GSH is an unnatural amino acid, which cannot be used in cellular metabolism. Consequently, the undigested D-GSH-modified UYTe accumulated in the cytoplasm and increased the UYTe vestigial in cell, leading to autophagy (**Figure S20-21**). Third, the accumulated D-GSH-modified UYTe further enhanced the oxidative stress in living cells (**Figure S22**). Moreover, the UYTe structure showed higher ROS production than UCNP-centered Au NP tetrahedron structure due to the strong plasmonic intensity of

YSNP, which induced enhanced autophagy degree after D-GSH modification of same concentration (**Figure S26-S28**).

3. UCL bio-imaging has been successfully used to visualize the autophagy effect *in vivo* as well as ATP concentration response.

We have added the control experiments to support our findings as your suggestion. As compared with the chiral UYTe assembly, the building block of single Au NP or UCNP showed extremely weak autophagy inducing ability even after D-GSH modification, which strongly demonstrated that the proposed chiral UYTe assembly was indispensable for high degree autophagy-inducing (**Figure S18-S19**). Furthermore, we replaced the YSNP with Au nanoparticles and assembled UCNP-Centered Au NP tetrahedron. Through detailed comparison, we found that the chiral UYTe assembly still had the higher autophagy-inducing ability than UCNP-Centered Au NP tetrahedron (**Figure S26-S28**). It was due to that the YSNP in UYTe assembly have the stronger ROS production which produce enhanced oxidative stress in living cells. More importantly, through thymine treatment, the ATP-related cell cycle was blocked, after incubated with this chiral UYTe assembly we proved that the ATP generation was closely connected with the autophagy degree (**Figure S23**). Therefore, the induced autophagy degree could be real-time monitored through UCL intensity of chiral UYTe assembly in living cell and *in vivo*. From these experiments, we proved that the chiral UYTe assembly was the unique structure for both high autophagy-inducing and detection in living cells.

Detailed responses are as follows.

Scheme 1. (a) the self-assembly process of UYTe and the detection principle for autophagy and ATP (b) the induced autophagy and corresponding ATP quantitative detection both in living cell and in mice.

Figure S20. The bio-TEM images of MCF-7 cells (5×10^7) treated by (a) D-GSH modified UYTe (40 nM), and (b) L-GSH modified UYTe (40 nM) for 12h.

Figure S21. The intracellular bio-distribution of Au amounts after 24 h treatment of D- / L-GSH modified UYTe were measured by ICP-MS.

Figure S22. The ROS production of MCF-7 cells (5×10^7) treated by PBS, UYTe (40 nM) in PBS, L-GSH modified UYTe (40 nM), and D-GSH modified UYTe (40 nM) for 12h measured by H2DCFDA (Invitrogen™) (GSH concentration is 5 μ M).

Figure S26. The confocal images of MCF-7 cells (5×10^7) treated by UCNP-Centered Au NP tetrahedron (40 nM) with PBS, L-GSH modified UCNP-Centered Au NP tetrahedron (40 nM) and D-GSH modified UCNP-Centered Au NP tetrahedron (40 nM)12h. Scale bar 20 μ m.

Figure S27. The bio-TEM images of MCF-7 cells (5×10^7) treated by (a) D-GSH modified UCNP-Centered Au NP tetrahedron (40 nM), (b) L-GSH modified UCNP-Centered Au NP tetrahedron (40 nM) 12h.

Figure S28. The ROS production of MCF-7 cells (5×10^7) treated by PBS, UCNP-Centered Au NP tetrahedron (40 nM) in PBS, L-GSH modified UCNP-Centered Au NP tetrahedron (40 nM), and D-GSH modified UCNP-Centered Au NP tetrahedron (40 nM) for 12h measured by H2DCFDA (Invitrogen™) (GSH concentration is 5 μ M).

Figure S18. The confocal images of MCF-7 cells (5×10^7) treated by D-GSH-modified gold nanoparticle (40 nM) for 12h. Scale bar 20 μm .

Figure S19. The confocal images of MCF-7 cells (5×10^7) treated by D-GSH-modified UCNP (40 nM) for 12h. Scale bar 20 μm .

Figure S23. The confocal images of MCF-7 cells (5×10^7) treated by UYTe (40 nM) with PBS, L-GSH modified UYTe (40 nM), UYTe (40 nM) with Rapamycin (autophagy inducer 10 μ M) and D-GSH modified UYTe (40 nM) 12h after thymine treatment. Scale bar 20 μ m.

Suggested improvements:

1. In this manuscript, my understanding is that the authors want to present two arguments: one is that the D-GSH-modified UYTe can induce autophagy; the other is the induction of autophagy can be monitored by ATP-released UCL fluorescence. However, one should bear in mind that many types of materials have been reported to induce autophagy such as gold nanoparticles (Biomaterials, 2010, 31, 5996-6003) and upconversion nanoparticles (J. Am. Chem. Soc. 2015, 137, 6550–6558). Would these materials have the same effect as D-GSH-modified UYTe upon coating with GSH?

Reply 1-1: Thanks for your constructive comment. As you mentioned that the gold

nanoparticles (*Biomaterials*, 2010, 31, 5996-6003) could also induce autophagy concomitant with oxidative stress. However, there are only focused on the single plasmonic gold nanoparticle. In spite of this, the link between autophagy and chiral plasmonic assembly have not been well established. Moreover, the corresponding production of ATP content could not be monitored due to the single gold nanoparticle. Also, the upconversion nanoparticles was found that could induced cell death through two distinct cell death pathways, autophagy and apoptosis (*J. Am. Chem. Soc.* 2015, 137, 6550–6558). The mechanism of autophagy in that paper was due to the deprivation of ATP by ligand-free UCNP, which induced the high cytotoxicity associated with lanthanide-doped nanoparticles. As compared with our manuscript, they just investigated the mechanism of autophagy inducing ability, the intracellular autophagy-inducing activity could not be directly measured and monitored through single UCNP. Moreover, we also studied the autophagy inducing ability of D-GSH-modified gold nanoparticle (20 ± 2.4 nm). As illustrated in the **Figure S18**, the cell treated by D-GSH-modified gold nanoparticle showed weak autophagy degree due to the low plasmonic resonance and small size of single NP. Also, there was no UCL signal for real-time monitor of autophagy.

We also tested the autophagy-inducing activity of D-GSH-modified UCNP (**Figure S19**). The UCL signal was lighting up in living cell which indicated that the UCNP was successfully entered cells. However, the autophagy kit only showed little intensity after 12 h incubation of D-GSH-modified UCNP. It was due to that the UCNP in our experiment are coated through the PEG layer, which could increase the bio-compatibility in living cell (*ACS nano*, 2013, 7(8): 7227-7240.). And in this case, UCL signal could not be used to monitor the autophagy-inducing activity in living cell. The corresponding changes are highlighted in red.

Figure S18. The confocal images of MCF-7 cells (5×10^7) treated by D-GSH-modified gold nanoparticle (40 nM) for 12h. Scale bar 20 μm .

Figure S19. The confocal images of MCF-7 cells (5×10^7) treated by D-GSH-modified UCNP (40 nM) for 12h. Scale bar 20 μm .

2. Other researchers have examined the effects of QDs, coated with different chiral forms of GSH, on induction of autophagy (*Angew. Chem. Int. Ed.* 2011, 50, 5860–5864). The chirality-dependent activation of autophagy was. The main difference is that in the published work the L-GSH has proven more effective than D-GSH.

Reply 1-2: Thank you for your nice suggestion and we are sorry for the unclear expression. As you mentioned that the effects of CdTe QDs capped with different chiral GSH on its cytotoxicity and autophagy-inducing ability were reported (*Angew.*

Chem. Int. Ed. 2011, 50, 5860–5864). The activation of autophagy was chirality-dependent, which the L-GSH-QDs showed more effective autophagy-inducing ability than D-GSH-QDs. Actually, the mechanism was due to that the biologically inactive D-GSH coating acted as “inert” surface which prevent the interaction between CdTe core and cellular component. While the L-GSH could be applied in cellular activity, the naked CdTe QDs showed much more cytotoxicity than D-GSH-coated QDs.

In our work, the D-GSH-modified UYTe showed significant autophagy-inducing ability than L-GSH coating. The reasons are as follows, first, the increased size of the particles after assembly could increase the accumulation of autophagosomes (*ACS nano*, 2011, 5(11): 8629-8639).

Second, D-GSH is an unnatural amino acid, which cannot be used in cellular metabolism. Consequently, the undigested D-GSH-modified UYTe accumulated in the cytoplasm, leading to autophagy. Through the bio-TEM images of MCF-7 cells (5×10^7) after treated by L-GSH modified UYTe (40 nM) or D-GSH modified UYTe (40 nM) for 12h (**Figure S20**), the extensive and huge aggregation of D-GSH modified UYTe in living cell was observed. While the L-GSH modified UYTe only showed limited aggregation in endocytosis vesicles. We also checked the Au content in living cells after incubated with L-GSH modified UYTe (40 nM) or D-GSH modified UYTe (40 nM) for 24h (**Figure S21**), it clearly showed that the UYTe after D-GSH coating have the higher residual content when the incubation time was more than 12h.

Third, we also investigated the ROS production of MCF-7 cells (5×10^7) after treated by PBS or UYTe (40 nM) in PBS or L-GSH modified UYTe (40 nM) or D-GSH modified UYTe (40 nM) for 12h through the ROS-sensitive fluorescent dye H2DCFDA (Invitrogen™). It showed that the oxidative stress in living cell under D-GSH modified UYTe treatment was higher than L-GSH modified UYTe and UYTe treatment, demonstrating that the autophagy inducing ability was related with the oxidative stress (**Figure S22**).

Our research has several novelties and is different from the previous reports. The corresponding changes are highlighted in red.

Figure S20. The bio-TEM images of MCF-7 cells (5×10^7) treated by (a) D-GSH modified UYTe (40 nM), and (b) L-GSH modified UYTe (40 nM) for 12h.

Figure S21. The intracellular bio-distribution of Au amounts after 24h treatment of D- / L-GSH modified UYTe were measured by ICP-MS.

Figure S22. The ROS production of MCF-7 cells (5×10^7) treated by PBS, UYTe (40 nM) in PBS, L-GSH modified UYTe (40 nM), and D-GSH modified UYTe (40 nM) for 12h measured by H2DCFDA (Invitrogen™) (GSH concentration is 5 μ M).

3. It is true that ATP can release UCL fluorescence, but I do not think the increment of ATP are only caused by autophagy. ATP provides energy for cell growth, proliferation, and even apoptosis. Thus, I do not think UCL fluorescence can be directly linked to autophagy.

Reply 1-3: Thanks for your constructive comment. In order to prevent the

interference from the ATP-related cellular activity, the MCF-7 cells were first treated by the thymine which the cell cycle was blocked. Then the chirality-dependent autophagy-inducing activity was monitored through the production of ATP. Through the confocal imaging of MCF-7 cells (5×10^7) treated by D-GSH-modified UYTe (40 nM) for 12h (**Figure S23**). The result illustrated that the autophagy could also be induced in living cell after thymine treatment. Meanwhile, although the cell cycle was blocked, the UCL intensity could still be excited, indicating that the ATP production was only induced by the autophagy in living cell. This was further proved from the positive-control of the cell incubated with UYTe (40 nM) and Rapamycin. It showed that the ATP production was increased as autophagy intensity enhanced. However, there was weak autophagy intensity after UYTe (40 nM) with PBS and L-GSH modified UYTe (40 nM) treatment. Also, the ATP production was also weak as illustrated from the UCL intensity. This results strongly proved that the UCL intensity of ATP production was directly connected with the autophagy degree. The corresponding changes are highlighted in red.

Figure S23. The confocal images of MCF-7 cells (5×10^7) treated by UYTe (40 nM) with PBS, L-GSH modified UYTe (40 nM), UYTe (40 nM) with Rapamycin (autophagy inducer 10 μ M) and D-GSH modified UYTe (40 nM) 12h after thymine treatment. Scale bar 20 μ m.

4. It puzzles me that the authors use Au@Ag@Au instead of just Au nanoparticles to construct UYTe structures (Figure 1a).

Reply 1-4: Thanks for your constructive comment. We have added the tetrahedral structure of Au nanoparticles (25.4 ± 2.2 nm) which were also centralized with UCNPs. As illustrated in the TEM images, the UCNP-centered Au NP tetrahedron structure was prepared in high yield (**Figure S24**). The bare assembly without GSH modification showed positive CD signal about 13 ± 1.1 mdeg. After coupled with L-GSH the CD intensity of UCNP-centered Au NP tetrahedron structure was

significantly enhanced to 30 ± 2.3 mdeg, while with the D-GSH modification, the assembly displayed a negative signal of -2.4 ± 0.5 mdeg (**Figure S25**). The generation of CD signal in UCNP-centered Au NP tetrahedron structure can be explained as follows. First, the tetrahedron assembly displayed a positive CD signal arising from its chiral tetrahedron shape. Second, with GSH modification, the Au NP could generate an intensive plasmon-induced chiral signal from the chiral GSH. As compared with UCNP-centered yolk-shell nanoparticles tetrahedron, the CD signal of UCNP-centered Au NP tetrahedron structure before and after GSH modification was lower. It was due to that the anisotropy of shell structure was higher than Au NPs and the shell structure have the strong plasmonic enhancement, which could induce the high CD signal after GSH modification (*Advanced Materials*, 2017, 29(18): 1606864.).

Also, we studied the autophagy-inducing ability of UCNP-centered Au NP tetrahedron structure before and after GSH modification (**Figure S26**). From the confocal images, it showed that the UCNP-centered Au NP tetrahedron structure without GSH modification showed weak autophagy-inducing ability and the ATP production could also be monitored from the UCL intensity. Also, the D-GSH modified UCNP-centered Au NP tetrahedron structure displayed higher autophagy inducing ability than L-GSH modification. Notably, although both assemblies were modified by GSH, the autophagy inducing ability of D-GSH modified UCNP-centered Au NP tetrahedron structure was lower than the D-GSH modified UYTe assembly as compared with **Figure 3b**. From the bio-TEM of cells treated by D- or L- GSH modified UCNP-Centered Au NP tetrahedron for 12 h, it also showed the high degree of aggregation after D-GSH modification (**Figure S27**). However, as compared with D-GSH modified UYTe assembly, the aggregation degree was still weak (**Figure S20**). Furthermore, the ROS production of UCNP-centered Au NP tetrahedron structure before and after GSH modification was measured (**Figure S28**). It showed that the D-GSH modified UCNP-centered Au NP tetrahedron structure has

lower ROS production than D-GSH modified UYTe in cells after 12 h incubation, which cause the weak autophagy-inducing ability in living cell (**Figure S22**). The reason was mainly due to that the shell structure with the plasmonic intensity has the high ROS production ability than Au NP, which were proved by previous report (*Advanced Materials*, 2017, 29(18): 1606864.).

Figure S24. The TEM images of UCNP-centered Au NP tetrahedron structure.

Figure S25. The CD spectrum of UCNP-centered Au NP tetrahedron structure before and after D- or L- GSH modification.

Figure S26. The confocal images of MCF-7 cells (5×10^7) treated by UCNP-Centered Au NP tetrahedron (40 nM) with PBS, L-GSH modified UCNP-Centered Au NP tetrahedron (40 nM) and D-GSH modified UCNP-Centered Au NP tetrahedron (40 nM)12h. Scale bar 20 μ m.

Figure 3b. The confocal images of MCF-7 cells (5×10^7) treated by UYTe (40 nM) with PBS, L-GSH modified UYTe (40 nM), UYTe (40 nM) with Rapamycin (autophagy inducer 10 μ M) and D-GSH modified UYTe (40 nM) for 12h and detected by the Premo™ Autophagy Sensor LC3B-RFP, BacMam 2.0 (Thermo Fisher). Scale bar 20 μ m.

Figure S27. The bio-TEM images of MCF-7 cells (5×10^7) treated by (a) D-GSH modified UCNP-Centered Au NP tetrahedron (40 nM), (b) L-GSH modified UCNP-Centered Au NP tetrahedron (40 nM) 12h.

Figure S20. The bio-TEM images of MCF-7 cells (5×10^7) treated by (a) D-GSH modified UYTe (40 nM), and (b) L-GSH modified UYTe (40 nM) for 12h.

Figure S28. The ROS production of MCF-7 cells (5×10^7) treated by PBS, UCNP-Centered Au NP tetrahedron (40 nM) in PBS, L-GSH modified UCNP-Centered Au NP tetrahedron (40 nM), and D-GSH modified UCNP-Centered Au NP tetrahedron (40 nM) for 12h measured by H2DCFDA (Invitrogen™) (GSH concentration is 5 μ M).

Figure S22. The ROS production of MCF-7 cells (5×10^7) treated by PBS, UYTe (40 nM) in PBS, L-GSH modified UYTe (40 nM), and D-GSH modified UYTe (40 nM) for 12h measured by H2DCFDA (Invitrogen™) (GSH concentration is 5 μ M).

5. Page 4, Line 85, “one of YSNPs was modified with peptide, ATG4B.” and Page 4, Line 90, “When it encountered ATG4B” is contradiction. What is ATG4B, an enzyme or a peptide?

Reply 1-5: Thank you for your comment and we are sorry for the mistake. ATG4B is one of the most important autophagy-related cysteine proteases and has been utilized as a potential autophagic biomarker. FGFT (sequence: Cys-Phe-Gly-Phe-Thr) is a responsive peptide sequence that could be hydrolyzed by the autophagy-related proteases, ATG4B (*Autophagy*, 2015, 11(2): 403-415; *Autophagy*, 2011, 7(9): 1052-1062.). As illustrated in **Scheme 1**, YSNPs dimer was formed by DNA self-assembly. Meanwhile, one of YSNPs of dimer structure was modified with Cysteine-modified linker peptide, FGFT. We have revised the sentence in the new revision and the relevant parts are highlighted in red.

As illustrated in **Scheme 1**, YSNPs dimer was formed by DNA self-assembly. Meanwhile, one of YSNPs was modified with responsive linker peptide, FGFT (sequence: Cys-Phe-Gly-Phe-Thr), which could be hydrolyzed by the autophagic biomarker of ATG4B.

Scheme 1. (a) the self-assembly process of UYTe and the detection principle for autophagy and ATP (b) the induced autophagy and corresponding ATP quantitative detection both in living cell and in mice.

6. There are quite many similar abbreviations throughout the manuscript.

Reply 1-6: Thank you for your comment and we are sorry for the unclear expression. We have revised the abbreviations in the new revision and the relevant parts are highlighted in red.

Figure 1. The TEM images of (a) YSNP dimer, (b) YSNP-YSNP-UCNP trimer, (c) UYTe. The corresponding EDX mapping images of UYTe. (e) The yield of YSNP dimer, YSNP-YSNP-UCNP trimer and UYTe during the self-assembly progress. (f) The electrophoresis image of AuNP, YSNPs, YSNP dimer, YSNP-YSNP-UCNP trimer and UYTe. (g) The corresponding 3D reconstruction cryo-TEM tomography

image of UYTe. (h) The SAXS spectrum of YSNPs, UYTe and **YSNP tetrahedrons**.

“sp” means single-particle, “di” and “tri” stand for assemblies consist of two and three-particle, respectively, “oth” means other assemblies.

Figure 2. The (a) CD and (b) UCL spectrum of UCNP, YSNPs, **YSNP dimer**, **YSNP-YSNP-UCNP trimer** and UYTe assemblies. The TEM images of (c) D-GSH and (d) L-GSH modified UYTe. The TEM images of (e) D-GSH and (f) L-GSH modified YSNP tetrahedrons. (g) The CD spectrum of D-/ L-GSH modified UYTe and **YSNP tetrahedrons**.

7. Figure 3C looks blurry. For improvement, authors can refer to Figure 2C in the

published paper (Angew Chem Int Ed 2011, 50, 5860–5864)

Reply 1-7: Thank you for your comment and we are sorry for the unclear expression. We have improved the bio-TEM images in **Figure 3C**. Moreover, the corresponding enlarged bio-TEM images of MCF-7 cells after different treatment were also provided in supporting information of new revision and highlighted in red (**Figure S29-S32**).

Figure 3C. The bio-TEM images of MCF-7 cells (5×10^7) treated by UYTe (40 nM) with PBS, L-GSH modified UYTe (40 nM), UYTe (40 nM) with Rapamycin (autophagy inducer 10 μ M) and D-GSH modified UYTe (40 nM) for 12h. The blue arrow indicates the autophagosomes and the white arrow means the autolysosomes.

Figure S29. Enlarged bio-TEM images of MCF-7 cells (5×10^7) treated by UYTe (40 nM) with Rapamycin (autophagy inducer 10 μ M) (40 nM) for 12h. The blue arrow indicates the autophagosomes and the white arrow means the autolysosomes.

Figure S30. Enlarged bio-TEM images of MCF-7 cells (5×10^7) treated by D-GSH modified UYTe (40 nM) for 12h. The blue arrow indicates the autophagosomes and the white arrow means the autolysosomes.

Figure S31. Enlarged bio-TEM images of MCF-7 cells (5×10^7) treated by L-GSH modified UYTe (40 nM) for 12h. The blue arrow indicates the autophagosomes and the white arrow means the autolysosomes.

Figure S32. Enlarged bio-TEM images of MCF-7 cells (5×10^7) treated by UYTe (40 nM) with PBS for 12h. The blue arrow indicates the autophagosomes and the white arrow means the autolysosomes.

8. What is FGFT? I cannot find the description in text.

Reply 1-8: Thank you for your comment and we are sorry for the unclear expression. FGFT (sequence: Cys-Phe-Gly-Phe-Thr) is a responsive peptide sequence that could

be hydrolyzed by the autophagy-related proteases, ATG4B (*Autophagy*, 2015, 11(2): 403-415; *Autophagy*, 2011, 7(9): 1052-1062.). As illustrated in **Scheme 1**, YSNPs dimer was formed by DNA self-assembly. A responsive linker peptide, FGFT, which connected between DNA and one of YSNPs were prepared. Meanwhile, ATP aptamer sequence modified UCNP was hybridized with the other YSNP dimer and formed YSNP-YSNP-UCNP trimer structure. Finally, the dimer and trimer were combined into a UYTe structure by DNA complementary. When it encountered ATG4B, the specific cleavage of the FGFT peptide caused the disassembly of YSNP in one corner and a reduction in the CD signal, whereas the UCL intensity was restored by the activation of ATP production during autophagy. We have added the explanation of FGFT in the new revision and the relevant parts are highlighted in red.

Meanwhile, one of YSNPs was modified with responsive linker peptide, FGFT (sequence: Cys-Phe-Gly-Phe-Thr), which could be hydrolyzed by the autophagic biomarker of ATG4B.

Scheme 1. (a) the self-assembly process of UYTe and the detection principle for autophagy and ATP (b) the induced autophagy and corresponding ATP quantitative detection both in living cell and in mice.

Reviewer #2 (Remarks to the Author):

In the current manuscript, the authors come up with a novel idea of generating chiral, tetrahedral DNA programmed devices to induce and monitor Autophagy in cells and tissues. The authors explore the chiral properties of fluorescence responsive biomolecules like GSH and ATP as a reporter. They successfully couple the nanodevices with Autophagy sensing peptide (ATGB4), the recognition of which with cellular targets, triggers the disassembly of nanodevice leading to fluorescence readout corresponding to extent of Autophagy and the involvement of ATP in this cellular process.

However, the manuscript suffers from novelty and some major factors (see below) need a strong attention before the manuscript becomes suitable for publication in Nature Communications.

Reply: We thank the reviewer for his/her excellent and constructive comments. The novelty in our manuscript are as follows:

1. A chiral nanodevice made of yolk–shell nanoparticles tetrahedron centralized with UCNPs (UYTe) has been fabricated with the yield of $84.6\% \pm 1.7\%$ for first time.
2. It has been proved the chiral UYTe nanodevice showed chirality dependent autophagy-inducing ability and ATP production in living cell. The mechanism are as follows: first, the increased size of the particles after assembly could increase the accumulation of autophagosomes. Second, D-GSH is an unnatural amino acid, which cannot be used in cellular metabolism. Consequently, the undigested D-GSH-modified UYTe accumulated in the cytoplasm and increased the UYTe vestigial in cell, leading to autophagy (**Figure S20-21**). Third, the accumulated D-GSH-modified UYTe further enhanced the oxidative stress in living cells (**Figure S22**). Moreover, the UYTe structure showed higher ROS production than UCNP-centered Au NP tetrahedron structure, which induced enhanced autophagy degree after D-GSH modification of same concentration (**Figure S26-S28**).

3. UCL bio-imaging has been successfully used to visualize the autophagy effect in vivo as well as ATP concentration response.

We have investigated the time of the UYTe assembly entered the cell and the starting time to induce autophagy in detail. Moreover, to further proved this process, the enlarged confocal images and immuno-electron microscope images were provided in new revision. Detailed responses are as follows.

Figure S20. The bio-TEM images of MCF-7 cells (5×10^7) treated by (a) D-GSH modified UYTe (40 nM), and (b) L-GSH modified UYTe (40 nM) for 12h.

Figure S21. The intracellular bio-distribution of Au amounts after 24h treatment of D- / L-GSH modified UYTe were measured by ICP-MS.

Figure S22. The ROS production of MCF-7 cells (5×10^7) treated by PBS, UYTe (40 nM) in PBS, L-GSH modified UYTe (40 nM), and D-GSH modified UYTe (40 nM) for 12h measured by H2DCFDA (Invitrogen™) (GSH concentration is 5 μ M).

Figure S26. The confocal images of MCF-7 cells (5×10^7) treated by UCNP-Centered Au NP tetrahedron (40 nM) with PBS, L-GSH modified UCNP-Centered Au NP tetrahedron (40 nM) and D-GSH modified UCNP-Centered Au NP tetrahedron (40 nM)12h. Scale bar 20 μ m.

Figure S27. The bio-TEM images of MCF-7 cells (5×10^7) treated by (a) D-GSH modified UCNP-Centered Au NP tetrahedron (40 nM), (b) L-GSH modified UCNP-Centered Au NP tetrahedron (40 nM) 12h.

Figure S28. The ROS production of MCF-7 cells (5×10^7) treated by PBS, UCNP-Centered Au NP tetrahedron (40 nM) in PBS, L-GSH modified UCNP-Centered Au NP tetrahedron (40 nM), and D-GSH modified UCNP-Centered Au NP tetrahedron (40 nM) for 12h measured by H2DCFDA (Invitrogen™) (GSH concentration is 5 μ M).

1. The concept is certainly good and innovative. However, the novelty of the idea is compromised since most of these things are already known in literature and published results (ref 22-28, 38-39). The main problem is that this device induces autophagy and then the authors could sense it. Is it possible to only sense the autophagy and not induce it. For example, if they take cells grown in normal FBS and the serum starved cells, which have high autophagy, and then use the same device without ATGB peptide, is it possible to still sense the autophagy. Mostly, autophagosomes are lysosomes or autophagolysosomes. And bulk of the recent literature on DNA nanodevices show that DNA devices post uptake are targeted to lysosomes. So in principle it should be possible to generate only a sensor and not the inducer.

Reply 2-1: Thanks for your constructive comment. We first assembled chiral nanodevice of yolk-shell nanoparticles tetrahedron centralized with UCNPs (UYTe), which has never been reported before. Then, through the GSH-modification, we found that the UYTe assembly could induce the autophagy intensity in living cell. Also, the intracellular ATP production during autophagy could be monitored. Although many

references as you mentioned have reported that the autophagy could be induced and detected by nanomaterial, there was no report on the autophagy inducing ability of chiral plasmonic assembly and the autophagy degree detection through autophagy related ATP production. In our work, the D-GSH-modified UYTe assembly showed significant autophagy-inducing ability than L-GSH coating. The mechanism is different from other reports, first, the increased size of the particles after assembly could increase the accumulation of autophagosomes (*ACS nano*, 2011, 5(11): 8629-8639).

Second, D-GSH is an unnatural amino acid, which cannot be used in cellular metabolism. Consequently, the undigested D-GSH-modified UYTe accumulated in the cytoplasm, leading to autophagy. Through the bio-TEM images of MCF-7 cells (5×10^7) after treated by L-GSH modified UYTe (40 nM) or D-GSH modified UYTe (40 nM) for 12h (**Figure S20**), the extensive and huge aggregation of D-GSH modified UYTe in living cell was observed. While the L-GSH modified UYTe only showed limited aggregation in endocytosis vesicles. We also checked the Au content in living cells after incubated with L-GSH modified UYTe (40 nM) or D-GSH modified UYTe (40 nM) for 24h (**Figure S21**), it clearly showed that the UYTe after D-GSH coating have the higher residual content when the incubation time was more than 12h.

Third, we also investigated the ROS production of MCF-7 cells (5×10^7) after treated by PBS or UYTe (40 nM) in PBS or L-GSH modified UYTe (40 nM) or D-GSH modified UYTe (40 nM) for 12h through the ROS-sensitive fluorescent dye H2DCFDA (Invitrogen™). It showed that the oxidative stress in living cell under D-GSH modified UYTe treatment was higher than L-GSH modified UYTe and UYTe treatment, demonstrating that the autophagy inducing ability was related with the oxidative stress (**Figure S22**).

So, there are many differences between our work and other researchers.

The proposed UYTe assembly could be used as only autophagy sensor and not induce it. As illustrated in **Figure 3a**, the UYTe assembly without GSH modification were

first used to quantitative detection the intracellular ATP. Before that, the cells were treated with an ATP inhibitor or inducer, respectively. The cells incubated with high-concentration ATP inhibitor produced only a limited UCL signal on confocal microscopic images, indicating the successful inhibition of ATP activity. However, when treated with the ATP inducer, the cells displayed distinctly enhanced UCL intensity compared with that in the PBS-treated group. This clearly demonstrates that the UCL signal of the UYTe assembly recovered as the intracellular ATP concentration increased. This result demonstrated that the UYTe structure could be used as intracellular ATP sensor.

Furthermore, we supplied the control group of the cell incubated in serum starve media. In order to detect the autophagy induced by cell starve, the MCF-7 cells (5×10^7) were incubated with UYTe without GSH modification (40 nM) for 12h in the serum starve media. The UYTe was acted as a sensor not an inducer. As illustrated in **Figure S33**, the cells showed high autophagy intensity after incubated in serum starve media. Simultaneously, the UCNP was illuminated during starve treatment, indicating that the production of ATP was increased as the autophagy induced. This result displayed strong evidence that the UYTe nanostructure without GSH modification could also be used for autophagy detection. Moreover, the ATP production is tightly connected with the autophagy. The corresponding changes are highlighted in red.

Figure S20. The bio-TEM images of MCF-7 cells (5×10^7) treated by (a) D-GSH modified UYTe (40 nM), and (b) L-GSH modified UYTe (40 nM) for 12h.

Figure S21. The intracellular bio-distribution of Au amounts after 24h treatment of

D- / L-GSH modified UYTe were measured by ICP-MS.

Figure S22. The ROS production of MCF-7 cells (5×10^7) treated by PBS, UYTe (40 nM) in PBS, L-GSH modified UYTe (40 nM), and D-GSH modified UYTe (40 nM) for 12h measured by H2DCFDA (Invitrogen™) (GSH concentration is 5 μ M).

Figure 3. (a) The UCL confocal images of MCF-7 cells (5×10^7) treated with (a1) $20 \mu\text{g mL}^{-1}$, (a2) $10 \mu\text{g mL}^{-1}$ oligomycin (ATP inhibitor), (a3) PBS, or treated with (a4) $50 \mu\text{g mL}^{-1}$, (a5) $100 \mu\text{g mL}^{-1}$ etoposide (ATP inducer), after incubated with UYTe (40 nM), respectively. Scale bar 20 μm .

Figure S33. The confocal images of MCF-7 cells (5×10^7) treated UYTe without GSH modification (40 nM) for 12 h incubated in the serum starve media. Scale bar 20 μ m.

2. DNA tetrahedron is definitely a chiroptical molecule, but not in this case. In this case, the chirality is induced only by D or L - GSH. So, the main focus should be on the chirality induced by ligand and not by DNA. This is actually evident from fig. 3B where the authors show that L and not D-GSH decorated DNA devices can induce autophagy.

Reply 2-2: Thank you for your comment and we are sorry for the unclear expression. In this manuscript, the chiral nanodevice made of yolk-shell nanoparticles tetrahedron (UYTe), centralized with UCNPs was fabricated. The UYTe assembly modified with different enantiomers of GSH. The D-GSH-modified UYTe showed significant autophagy-inducing ability because the enhanced oxidative stress and accumulation in living cell. The activation of autophagy resulted in the reduced intracellular CD intensity which originated from the disassembly of the structure. The intracellular ATP concentration was simultaneously enhanced in response to autophagy activity, which was quantitatively bio-imaged with the UCL signal of the UCNP that escaped from

UYTe in living cells.

In **Figure 2g**, before GSH modification, the UYTe structure only showed 22.6 ± 1.4 mdeg. Notably, the CD intensity of the UYTe nanostructure after L-GSH modification was 54.7 ± 1.5 mdeg while after D-GSH modification it was -3.97 ± 1.1 mdeg. These results indicating that the chiral origin of UYTe structure after GSH modification was from the chiral configuration of DNA assembly and plasmonic induced-chirality of GSH modification.

The focus in our manuscript was chirality induced autophagy by GSH ligand-modified UYTe assembly. So, the autophagy inducing ability of UYTe after 5 μ M GSH modification was studied in **Figure 3b**. Under this condition, the D-GSH modified UYTe assembly showed high autophagy inducing ability while the autophagy degree of L-GSH modified UYTe was weak after 12h treatment. Meanwhile, the ATP production were monitored from the UCL intensity, which showed that the ATP generation was increased as the autophagy degree enhanced. With the UYTe assembly, the autophagy degree could be detected through UCL intensity both in living cell and *in vivo*. The mechanism are as follows: first, the increased size of the particles after assembly could increase the accumulation of autophagosomes (*ACS nano*, 2011, 5(11): 8629-8639). Second, D-GSH is an unnatural amino acid, which cannot be used in cellular metabolism. Consequently, the undigested D-GSH-modified UYTe accumulated in the cytoplasm and increased the UYTe vestigial in cell, leading to autophagy (**Figure S20-21**). Third, the accumulated D-GSH-modified UYTe further enhanced the oxidative stress in living cells (**Figure S22**). Moreover, the UYTe structure showed higher ROS production than UCNP-centered Au NP tetrahedron structure, which induced enhanced autophagy degree after D-GSH modification of same concentration (**Figure S26-S28**).

The corresponding changes are highlighted in red.

Figure 2g The CD spectrum of D-/ L-GSH modified UYTe and YSNP tetrahedrons with 5 μ M GSH modification.

Figure 3b. The confocal images of MCF-7 cells (5×10^7) treated by UYTe (40 nM) with PBS, L-GSH modified UYTe (40 nM), UYTe (40 nM) with Rapamycin (autophagy inducer 10 μ M) and D-GSH modified UYTe (40 nM) for 12h and detected by the Premo™ Autophagy Sensor LC3B-RFP, BacMam 2.0 (Thermo Fisher). Scale bar 20 μ m.

Figure S20. The bio-TEM images of MCF-7 cells (5×10^7) treated by (a) D-GSH modified UYTe (40 nM), and (b) L-GSH modified UYTe (40 nM) for 12h.

Figure S21. The intracellular bio-distribution of Au amounts after 24h treatment of D- / L-GSH modified UYTe were measured by ICP-MS.

Figure S22. The ROS production of MCF-7 cells (5×10^7) treated by PBS, UYTe (40 nM) in PBS, L-GSH modified UYTe (40 nM), and D-GSH modified UYTe (40 nM) for 12h measured by H2DCFDA (Invitrogen™) (GSH concentration is 5 μ M).

Figure S26. The confocal images of MCF-7 cells (5×10^7) treated by UCNP-Centered Au NP tetrahedron (40 nM) with PBS, L-GSH modified UCNP-Centered Au NP tetrahedron (40 nM) and D-GSH modified UCNP-Centered Au NP tetrahedron (40 nM) 12h. Scale bar 20 μm .

Figure S27. The bio-TEM images of MCF-7 cells (5×10^7) treated by (a) D-GSH modified UCNP-Centered Au NP tetrahedron (40 nM), (b) L-GSH modified UCNP-Centered Au NP tetrahedron (40 nM) 12h.

Figure S28. The ROS production of MCF-7 cells (5×10^7) treated by PBS, UCNP-Centered Au NP tetrahedron (40 nM) in PBS, L-GSH modified UCNP-Centered Au NP tetrahedron (40 nM), and D-GSH modified UCNP-Centered Au NP tetrahedron (40 nM) for 12h measured by H2DCFDA (Invitrogen™) (GSH concentration is 5 μ M).

3. Page 4, Please introduce for the first time what is ATGB4. Now in fig. 1 there is no mention of ATGB4. Is it FGFT in the fig 1?

Reply 2-3: Thank you for your carefully comment and we are sorry for the unclear expression. ATG4B is one of the most important autophagy-related cysteine proteases and has been utilized as a potential autophagic biomarker. FGFT (sequence: Cys-Phe-Gly-Phe-Thr) is a responsive peptide sequence that could be hydrolyzed by the autophagy-related proteases, ATG4B (*Autophagy*, 2015, 11(2): 403-415; *Autophagy*, 2011, 7(9): 1052-1062.). As illustrated in **Scheme 1**, YSNPs dimer was formed by DNA self-assembly. A responsive linker peptide, FGFT, which connected between DNA and one of YSNPs were prepared. Meanwhile, ATP aptamer sequence modified UCNP was hybridized with the other YSNP dimer and formed YSNP-YSNP-UCNP trimer structure. Finally, the dimer and trimer were combined into a UYTe structure by DNA complementary. When it encountered ATG4B, the specific cleavage of the FGFT peptide caused the disassembly of YSNP in one corner

and a reduction in the CD signal, whereas the UCL intensity was restored by the activation of ATP production during autophagy. We have added the explanation of FGFT in the new revision and the relevant parts are highlighted in red.

Meanwhile, one of YSNPs was modified with responsive linker peptide, FGFT (sequence: Cys-Phe-Gly-Phe-Thr), which could be hydrolyzed by the autophagic biomarker of ATG4B.

Scheme 1. (a) the self-assembly process of UYTe and the detection principle for autophagy and ATP (b) the induced autophagy and corresponding ATP quantitative detection both in living cell and in mice.

4. Page 5, 10 and other places: In situ / in vitro should be replaced by in cells or living cells.

Reply 2-4: Thanks for your careful reading of our paper and nice advice. We have revised the corresponding phrase in main text and marked in red.

The intracellular ATP concentration was simultaneously enhanced in response to autophagy activity, which was quantitatively bio-imaged with the upconversion luminescence (UCL) signal of the UCNP that escaped from UYTe **in living cells**.

This could rapidly and accurately monitor the autophagic state **in living cells** in real time.

The production of ATP was enhanced with the induction of autophagy, which triggered an increase in the intracellular UCL intensity **in living cells**.

After the assembly was characterized, we tested the feasibility of autophagy and ATP monitoring **in solution** with the UYTe nanostructure without GSH modification.

However, the cells treated with the L-GSH-coated UYTe assembly displayed low UCL intensity and an unchanged CD signal as **in solution**.

These results confirm the strong autophagy-inducing ability of the D-GSH-modified UYTe structure both **in solution** and *in vivo*, which is mainly attributable to its intracellular accumulation.

5. Fig. 1: Is there a control on number of FGFT peptides per gold nanoparticles or number of GSH? Ideally it should be one ligand, one nanoparticle but I guess for the cellular processes, multifunctionalization might be needed.

Also, would have been excellent to see if different degrees of functionalization of device behave differently in cells. But that's just a suggestion.

Reply 2-5: Thanks for your careful reading of our paper and nice advice. The FGFT peptides for YSNPs was 5 μM with the molar ratio of YSNP: peptide in 1:500, which could cover the surface of YSNP (**Figure S6**). The CD signal of YSNP after 5 μM GSH modification showed the highest intensity, which were used for further

experiment. After assembled into UYTe structure, the CD intensity of L-GSH modified UYTe was 54.7 ± 1.5 mdeg while the D-GSH was -3.97 ± 1.1 mdeg (**Figure 2g**). Also, the autophagy inducing ability of UYTe after 5 μ M GSH modification was studied in **Figure 3b**. Under this condition, the D-GSH modified UYTe assembly showed high autophagy inducing ability while the autophagy degree of L-GSH modified UYTe was weak after 12h treatment.

Then, we measured the CD intensity of UYTe after 1 μ M GSH modification. The CD intensity of L-GSH modified UYTe was 38.5 ± 2.5 mdeg while the D-GSH was -1.3 ± 0.4 mdeg, which the absolute value of CD intensity was smaller than the UYTe structure after 5 μ M GSH modification (**Figure S34**). Also, the autophagy inducing ability of UYTe after 1 μ M GSH modification was observed from confocal images. As displayed in **Figure S35**, the D-GSH modified UYTe assembly showed higher autophagy inducing ability than L-GSH modified UYTe after 12 h treatment. Notably, UYTe assembly with 5 μ M GSH modification displayed higher autophagy and UCL intensity than the UYTe assembly with 1 μ M GSH modification, which indicated that the autophagy inducing ability was enhanced as the absolute value of CD intensity increased. To further proved this phenomenon, the ROS production after UYTe assembly treatment of different concentration of GSH modification were measured. As compared with the UYTe assembly with 5 μ M GSH modification, the UYTe assembly with 1 μ M GSH modification showed lower ROS production due to the less concentration of the undigested D-GSH on UYTe in living cell, which induced the different degree of autophagy (**Figure S22 and S36**).

The number of FGFT peptide that are conjugated to the NP can be measured using the following procedure:

- (1) FGFT peptide was labeled with FAM fluorescent dyes during synthesis.
- (2) Then, a linear calibration curve was determined that relates to the concentration of a fluorescently labeled FGFT peptide and the fluorescence intensity.
- (3) After attachment of FAM labeled peptides to YSNP with a peptides-to-NP ratio of 500:1, centrifuge YSNP and measure fluorescence signal from the supernatant at the

490 nm excitation and 518 nm emission wavelengths of the FAM that were used to tag FGFT peptide.

(4) Use the calibration curves to determine the concentration of FGFT peptide in the supernatant. Subtract these values from the initial concentrations to obtain the amount of FGFT peptide that is attached to YSNP.

Figure S6. The CD spectra of YSNPs modified with different concentration of D-/ L-GSH.

Figure 2g The CD spectrum of D-/ L-GSH modified UYTe and YSNP tetrahedrons with 5 μ M GSH modification.

Figure S34. The CD spectrum of D-/ L-GSH modified UYTe with 1 μM GSH modification.

Figure S22. The ROS production of MCF-7 cells (5×10^7) treated by PBS, UYTe (40 nM) in PBS, L-GSH modified UYTe (40 nM), and D-GSH modified UYTe (40 nM) for 12h measured by H2DCFDA (Invitrogen™) (GSH concentration is 5 μM).

Figure S36. The ROS production of MCF-7 cells (5×10^7) treated by PBS, L-GSH ($1 \mu\text{M}$) modified UYTe (40 nM), and D-GSH ($1 \mu\text{M}$) modified UYTe (40 nM) for 12h measured by H2DCFDA (Invitrogen™).

Figure S35. The confocal images of MCF-7 cells (5×10^7) treated by L-GSH ($1 \mu\text{M}$) modified UYTe (40 nM) and D-GSH ($1 \mu\text{M}$) modified UYTe (40 nM) for 12h and detected by the Premo™ Autophagy Sensor LC3B-RFP, BacMam 2.0 (Thermo Fisher). Scale bar 20 μm .

Figure 3b. The confocal images of MCF-7 cells (5×10^7) treated by UYTe (40 nM) with PBS, L-GSH modified UYTe (40 nM), UYTe (40 nM) with Rapamycin (autophagy inducer 10 μ M) and D-GSH modified UYTe (40 nM) for 12h and detected by the Premo™ Autophagy Sensor LC3B-RFP, BacMam 2.0 (Thermo Fisher). Scale bar 20 μ m.

6. Page 6: Fig 1d is not mentioned in the fig legend later. please check.

Reply 2-6: Thanks for your comment and we are sorry for the negligence. We have revised the corresponding part in main text and marked in red.

Figure 1. (d) The corresponding EDX mapping images of UYTe.

7. Page 7: The size of the particles: Around 70 nm. Now, if we see any endocytic process like clathrin mediated, non-clathrin, caveolae,... in all the mechanisms the maximum size of the particles which gets endocytosed is not bigger than 50 nm. Thus, after 50 nm most of the particles get phagocytosed and endocytosed. The authors should do different times of incubation of these nanodevices to cells starting from few seconds, to mins to hours. Of course the autophagy will begin only after 30-40 mins (as against what authors show in Supplementary fig. S17 for 12 h), but it will give clue to authors how these devices enter into the cells and when they start to induce autophagy.

Reply 2-7: Thanks for your careful reading of our paper and nice advice. According to your suggestion, we first incubated the MCF-7 cells with D-GSH modified UYTe assembly. Then, the cell after different incubation time were collect and observed the autophagy-inducing ability through confocal images (**Figure S37**). When the incubation time was only 1 min, there was no any signal in cells indicating that the time was too short for assembly to enter the cell (*Nat. comm.*, 2017, 8 (1): 1847.). When the incubation time was increased to 30 min, there was a weak autophagy intensity demonstrating that the assembly began to induce autophagy when they

entered the cell after 30 min. The corresponding ATP production was also monitored, which showed the weak UCL signal as the autophagy happened. The way of the UYTe enter the cells was direct through membrane transport due to the use of cell-penetrating peptides (TAT) attached to the surface of assembly (*ACS nano*, 2011, 5(6): 5195-5201.). Furthermore, as the incubation time increased to 60 min, the autophagy intensity was enhanced. Also, the corresponding ATP intensity was increased as the autophagy degree. If the incubation time was increased to 2h or 6h, the autophagy degree and ATP production were extensively enhanced. The corresponding changes are highlighted in red.

Figure S37. The confocal images of MCF-7 cells (5×10^7) treated by D-GSH-modified UYTe (40 nM) for different times. Scale bar 20 μm .

8. Page 11: the authors claim that autophagy is dependent on ATP in cells. Please cite the reference. Without the reference, it is difficult to accept this claim.

Reply 2-8: Thanks for your comment and we are sorry for the negligence. We have already cited the reference that autophagy is dependent on ATP in living cells. It proved that the enhanced autophagic capacity and autophagosome correlated with secretion of ATP. We also added more reference to confirm this point of view and the corresponding part in main text and marked in red.

47. Martin S, *et al.* An autophagy-driven pathway of ATP secretion supports the aggressive phenotype of BRAF(V600E) inhibitor-resistant metastatic melanoma cells. *Autophagy* **13**, 1512-1527 (2017).
48. Wang YD, *et al.* Autophagy-dependent ATP release from dying cells via lysosomal exocytosis. *Autophagy* **9**, 1624-1625 (2013).
49. Li FJ, *et al.* ATP-driven and AMPK-independent autophagy in an early branching eukaryotic parasite. *Autophagy* **13**, 715-729 (2017).

9. The TEM images are very nice and self explanatory. However, one caution - are there some standards to show that the vesicles are autophagosomes or autolysosomes. Like some marker antibody coupled to gold nanoparticles used as markers for these organelles.

Reply 2-9: Thank you for your comment and we are sorry for the unclear expression. We have added the immuno-electron microscope images as your suggestion. MCF-7 cells were incubated with D- or L- GSH modified UYTe assembly for 12h, respectively. After that, the cell slice in paraformaldehyde were immersed in LC3-II antibody and the unreacted antibody were removed by PBS. Then, the second antibody of sheep Anti-Mouse IgG on AuNP was further incubated with the cell slice, which could target the auto autophagic membranes. As illustrated in bio-TEM of **Figure S38**, the auto autophagic membranes could be marked by Au NPs. The corresponding changes are highlighted in red.

Figure S38. Immuno-electron microscope images of the autophagic membranes in MCF-7 cells targeted by LC3-II antibody after (a) D-GSH modified UYTe (b) L-GSH modified UYTe.

10. Fig. 4 and 3: I suggest that authors also use LC3 antibody in fig. 3 where they take only cells and show that indeed the levels of autophagy has increased.

Also, please zoom in some of the regions of cells to show that indeed the staining shows punctate structures of autophagosomes and not just cytosolic or non-specific cellular staining.

Reply 2-10: Thank you for your comment and we are sorry for the unclear expression. The cells in Fig.3b were treated by UYTe (40 nM) with PBS, L-GSH modified UYTe

(40 nM), UYTe (40 nM) with Rapamycin (autophagy inducer 10 μ M) and D-GSH modified UYTe (40 nM) for 12h and detected by the LC3-II antibody in Premo™ Autophagy Sensor LC3B-RFP, BacMam 2.0 (Thermo Fisher). As illustrated in **Figure 3b**, the LC3-II protein could be selective targeted by the kit and labeled in green color. We presented the zoom in images of cells after different treatment. The cells treated with the D-GSH-coated UYTe structure produced a stronger green signal than other treatment, which was come from the LC3-II protein during autophagy. However, the L-GSH-coated UYTe assembly and the unmodified UYTe assembly produced only limited signals. Meanwhile, the ATP concentration was also monitored during this autophagy activity, which showed that the ATP production was increased as the degree of autophagy increased. Moreover, the ROS production after D-GSH-coated UYTe in living cell was studied, which demonstrated that the autophagy induced by the D-GSH-modified UYTe further enhanced the oxidative stress in living cells (**Figure S22**). These results indicating that the signal in confocal were come from the chiral plasmonic nanoassembly dependent autophagy in living cells not just cytosolic or non-specific cellular staining. The corresponding changes are highlighted in red.

Figure 3b. The confocal images of MCF-7 cells (5×10^7) treated by UYTe (40 nM) with PBS, L-GSH modified UYTe (40 nM), UYTe (40 nM) with Rapamycin (autophagy inducer 10 μ M) and D-GSH modified UYTe (40 nM) for 12h and detected by the Promo™ Autophagy Sensor LC3B-RFP, BacMam 2.0 (Thermo Fisher). Scale bar 20 μ m.

Figure S22. The ROS production of MCF-7 cells (5×10^7) treated by PBS, UYTe (40 nM) in PBS, L-GSH modified UYTe (40 nM), and D-GSH modified UYTe (40 nM) for 12h measured by H2DCFDA (Invitrogen™) (GSH concentration is 5 μ M).

11. Fig.1 - the schematic of the device operation is very complicated to understand. The authors should definitely try to simplify it.

Reply 2-11: Thanks for your careful reading of our paper and nice advice. We have revised the scheme figure. Hope it can clearly express the main idea of our work. As illustrated in **Scheme 1**, YSNPs dimer was formed by DNA self-assembly. A responsive linker peptide, FGFT (sequence: Cys-Phe-Gly-Phe-Thr), which connected between DNA and one of YSNPs were prepared. Meanwhile, ATP aptamer sequence modified UCNPs were hybridized with the other YSNP dimer and formed YSNP-YSNP-UCNP trimer structure. Finally, the dimer and trimer were combined into a UYTe structure by DNA complementary. When it encountered ATG4B, the specific cleavage of the FGFT peptide caused the disassembly of YSNP in one corner and a reduction in the CD signal, whereas the UCL intensity was restored by the activation of ATP production during autophagy (**Scheme 1a**). The prepared UYTe assembly was modified by the D-GSH or L-GSH, respectively. Then, the autophagy-inducing ability and corresponding ATP production of different chiral GSH modified UYTe were studied both in living cell and *in vivo* (**Scheme 1b**). The corresponding changes are highlighted in red.

Scheme 1. (a) the self-assembly process of UYTe and the detection principle for autophagy and ATP (b) the induced autophagy and corresponding ATP quantitative detection both in living cell and in mice.

12. Technical query - how did the authors measure UV and CD in cells? I could not find the procedure in methods or supplementary information.

Reply 5-12: Thanks for your comment and we are sorry for the negligence. We have added the corresponding procedure in supporting information and marked in red.

CD and UV-vis measurement

MCF-7 cells were seeded in a six-well plate with a density of 10^5 cells per well. The UYTe (40 nM) with PBS, L-GSH modified UYTe (40 nM), UYTe (40 nM) with Rapamycin (autophagy inducer 10 μ M) and D-GSH modified UYTe (40 nM) were co-cultivated with MCF-7 cells for 12 h at 37 °C, respectively. The cells were collected using standard trypsin-based lift-off protocol and washed with PBS three times. Then, the cells were re-dispersed in 200 μ L PBS and the chiroptical activity and UV-vis absorption were characterized by Chirascan (Applied Photophysics Ltd). The optical path length was 1 cm.

No issues with statistics and data analysis. All the ethics and rules for using cells lines and animal models seem to be followed by the authors.

Overall, good work but will need further revision before final decision.

Reply: Thanks for your hard work and comment. We have revised the manuscript as your suggestion.

All animal experiments conformed to the guidelines of the Chinese Animal Use and Care Committee.

REVIEWERS' COMMENTS:

Reviewer #1 (Remarks to the Author):

I have read the rebuttal letter and reviewed the changes to the manuscript. The technical content has significantly improved over the earlier version.

I believe the new information provided by the authors clears my previous concerns. I am happy to recommend publication of this manuscript.

Reviewer #2 (Remarks to the Author):

The authors have carefully addressed each of my claims supported by in-depth experiments and explanations. In my opinion, the revised manuscript is now suitable for publication in Nature Communications post editorial revision.